# Decoupling actin assembly from microtubule disassembly by TBC1D3C-mediated direct GEF-H1 activation

Yi Luan[1,2,3,4], Zhifeng Deng[1,2,3,4], Yutong Zhu[5], Lisi Dai[6,7], Yang Yang[1], Zongping Xia[1,2,3,4,8]

**Actin and microtubules are essential cytoskeletal components and coordinate their dynamics through multiple coupling and decoupling mechanisms. However, how actin and microtubule dynamics are decoupled remains incompletely understood. Here, we identified TBC1D3C as a new regulator that can decouple actin filament assembly from microtubule disassembly. We showed that TBC1D3C induces the release of GEF-H1 from microtubules into the cytosol without perturbing microtubule arrays, leading to RhoA activation and actin filament assembly. Mechanistically, we found that TBC1D3C directly binds to GEF-H1, disrupting its interaction with the Tctex-DIC-14-3-3 complex and thereby displacing GEF-H1 from microtubules independently of microtubule disassembly. Super-resolution microscopy and live-cell imaging further confirmed that TBC1D3C triggers GEF-H1 release and actin filament assembly while maintaining microtubule integrity. Therefore, our findings demonstrated that TBC1D3C functions as a direct GEF activator and a novel regulator in decoupling actin assembly from microtubule disassembly, providing new insights into cytoskeletal regulation.**

## Introduction

Actin and microtubules are two major cytoskeletal components that interact extensively and dynamically to mediate various biological processes (Drubin & Nelson, 1996; Li & Gundersen, 2008). Their dynamic assembly and disassembly are tightly coordinated by their physical interactions and various signaling molecules and cytoskeletal regulators (Ishizaki et al, 2001; Joo & Yamada, 2016).

During cell migration, actin filaments undergo rapid polymerization at the leading edge, whereas microtubules concurrently cycle through disassembly and assembly, ensuring coordinated and directional cell movement (Dogterom & Koenderink, 2019). During mitosis, microtubules assemble into the mitotic spindle for chromosome segregation, whereas actin filaments facilitate cytokinesis at the cleavage furrow (Dogterom & Koenderink, 2019). NDP52 has been identified as a key modulator that coordinates cortical actin–astral microtubule interactions, crucial for spindle orientation (Yu et al, 2019). During neuronal growth cone guidance, the dynamics of microtubules drive the forward translocation of the growth cone, whereas the dynamics of actin filaments guide its direction sensing (Etienne-Manneville, 2004). Their intricate coordination is mediated by various proteins. The microtubule-associated protein XMAP215 bridges interactions between microtubules and actin filament during growth cone growth (Slater et al, 2019). Similarly, the actin binding protein Fmn2 binds and stabilizes microtubules in growth cones, guiding them along actin bundles into chemosensory filopodia (Kundu et al, 2021). In addition, several microtubule plus-end tracking proteins (+TIPs), including ACF7/MACF1, APC, CLASPs, Ch-TOG, and NAV1, bind actin therefore crosslinking them together within growth cones (Zhou & Cohan, 2004; Sanchez-Soriano et al, 2009; van der Vaart et al, 2012; Marx et al, 2013; Sanchez-Huertas et al, 2020).

The dynamic interplay between microtubules and actin can be orchestrated by the Rho family GTPases. These GTPases function as molecular switches to mediate dynamic interactions between actin and microtubules in response to extracellular signals (Zhou et al, 2015). Microtubule assembly triggers Rac activation, promoting lamellipodium formation, whereas its disassembly activates Rho, leading to actin stress fiber formation (Liu et al, 1998). These GTPases are regulated by guanine nucleotide exchange factors (GEFs) and GTPase-activating proteins (GAPs) (Dogterom & Koenderink, 2019). These GEFs and GAPs regulate the cycling of GTPases between their inactive and active states, coordinating the dynamics of microtubules and actin in response to extracellular stimuli (Bos et al, 2007).

---

[1]Clinical Systems Biology Laboratories, Translational Medicine Center, The First Affiliated Hospital of Zhengzhou University, Zhengzhou, China  [2]Institute of Infection and Immunity, Henan Academy of Innovations in Medical Science, Zhengzhou, China  [3]Department of Neurology, The First Affiliated Hospital of Zhengzhou University, Zhengzhou, China  [4]NHC Key Laboratory of Prevention and Treatment of Cerebrovascular Diseases, The First Affiliated Hospital of Zhengzhou University, Zhengzhou, China  [5]Research and Development Center, Beijing, China  [6]Department of Pathology and Pathophysiology, and Department of Surgical Oncology of Second Affiliated Hospital, Zhejiang University School of Medicine, Hangzhou, China  [7]School of Basic Medical Sciences, Zhejiang University, Hangzhou, China  [8]Life Sciences Institute, Zhejiang University, Hangzhou, China

Correspondence: zxia2018@zzu.edu.cn

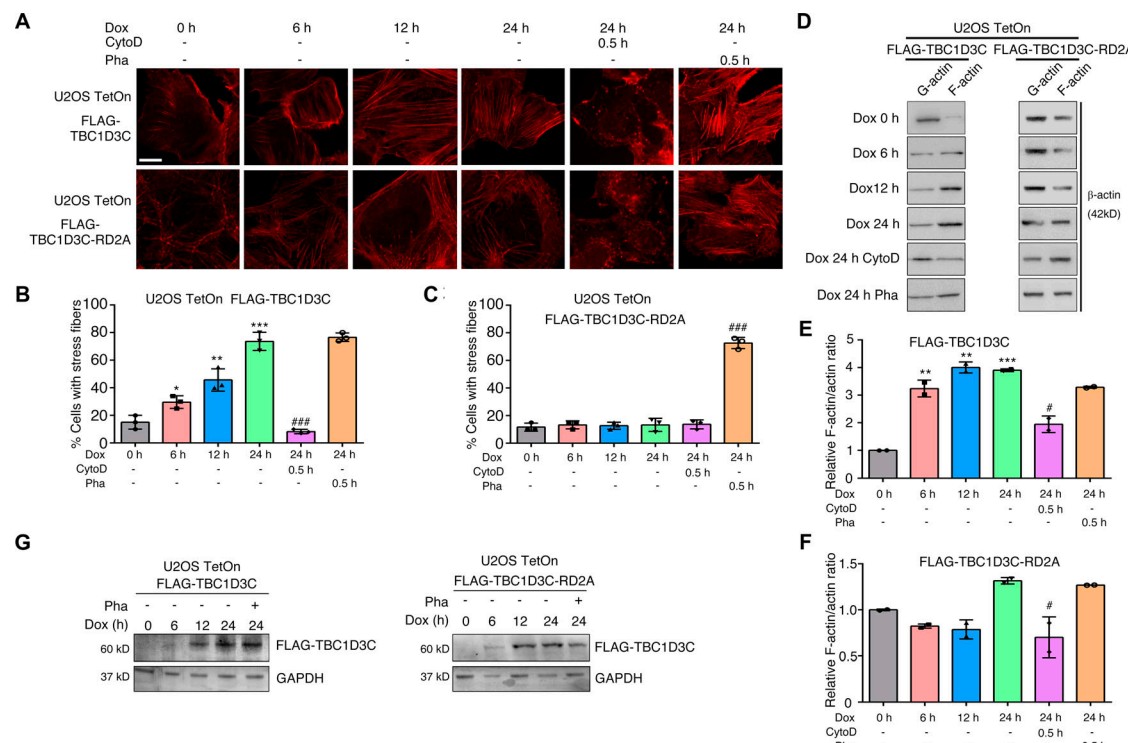

**Figure 1. TBC1D3C promotes F-actin filament formation.**
**(A, B, C)** TBC1D3C promotes actin filament (F-actin) formation; scale bar, 20 μm. **(B, C)** Quantification of F-actin in (A). U2OS *TBC1D3C* TetOn cells were grown on gelatin-coated cover glasses and treated with Dox, CytoD (3 μM, 0.5 h before fixation), or phalloidin (Pha, 200 nM, 0.5 h before fixation) for the indicated time, fixed, and subjected to immunofluorescence staining using phalloidin. **(D, E, F)** TBC1D3C induces F-actin formation. **(E, F)** Relative F-actin/actin ratio in (D). Cells were treated as indicated, and lysates were ultra-centrifuged to isolate assembled F-actin and monomer actin (G-actin) detected by Western blotting. **(G)** Verification of TBC1D3C and TBC1D3C-RD2A expression in the stable U2OS TetOn cell lines induced with Dox. *$P < 0.05$, **$P < 0.01$, ***$P < 0.001$ versus Cntrl group; #$P < 0.05$, ##$P < 0.01$, ###$P < 0.001$ versus Dox, 24 h.

GEF-H1 (ARHGEF2, also known as Lfc in mice) is a guanine nucleotide exchange factor that specifically regulates the spatio-temporal activation of Rho subfamily members RhoA, RhoB, and RhoC, which couple microtubule disassembly with actin stress fiber formation. GEF-H1 directly binds to microtubules, which inhibits its GEF activity (Ren et al, 1998; Chiang et al, 2014). However, upon microtubule depolymerization by external stimuli or microtubule-destabilizing agents, GEF-H1 is released and becomes activated, leading to the activation of RhoA and the assembly of actin stress fibers (Krendel et al, 2002). Various receptor ligands, such as lysophosphatidic acid (LPA) and thrombin, can trigger the activation of GEF-H1 and RhoA (Meiri et al, 2009; Kakiashvili et al, 2011). Despite its importance in cytoskeleton regulation, the molecular mechanisms that regulate GEF-H1 activity are not fully understood.

Although dynamic assembly and disassembly of actin and microtubules are often coupled, they can also function independently. The molecular mechanisms that underlie the decoupling of actin assembly and microtubule disassembly are still not fully understood. In this study, we identified TBC1D3C, a hominoid-specific protein with no known orthologs outside of the primate lineage (Wainszelbaum et al, 2008), as a novel regulator that can activate RhoA and promote actin filament formation without microtubule disassembly. Despite being structurally similar to GAPs for Rabs, TBC1D3C lacks the conserved residues essential for this activity (Bernards, 2003), making its biological function elusive. We

demonstrate that TBC1D3C binds directly to GEF-H1 and promotes its release from microtubules into the cytosol, where it can activate RhoA and stimulate actin filament assembly. Our findings suggest that TBC1D3C mediates the decoupling of actin assembly and microtubule disassembly through direct activation of GEF-H1, and shed new light on the complex crosstalk between the two major components of the cytoskeleton.

# Results

## TBC1D3C promotes F-actin filament formation

To investigate the role of TBC1D3C, we generated stable U2OS cell lines harboring either wild-type *TBC1D3C* or its mutant form *TBC1D3C*-RD2A (an R107A and D148A double-site mutant), allowing for controlled induction of TBC1D3C expression upon doxycycline (Dox) treatment (Fig 1G). Fluorescent phalloidin staining revealed prominent actin polymerization in cells after 12 and 24 h of TBC1D3C induction by Dox. However, treatment with the actin filament inhibitor cytochalasin D (CytoD) disrupted actin filaments and inhibited TBC1D3C-induced actin polymerization (Fig 1A–C). In contrast, the mutant form of TBC1D3C did not induce actin polymerization upon Dox treatment, and CytoD addition abrogated any bundled actin filaments (Fig 1A–C). CytoD is a cell-permeable fungal

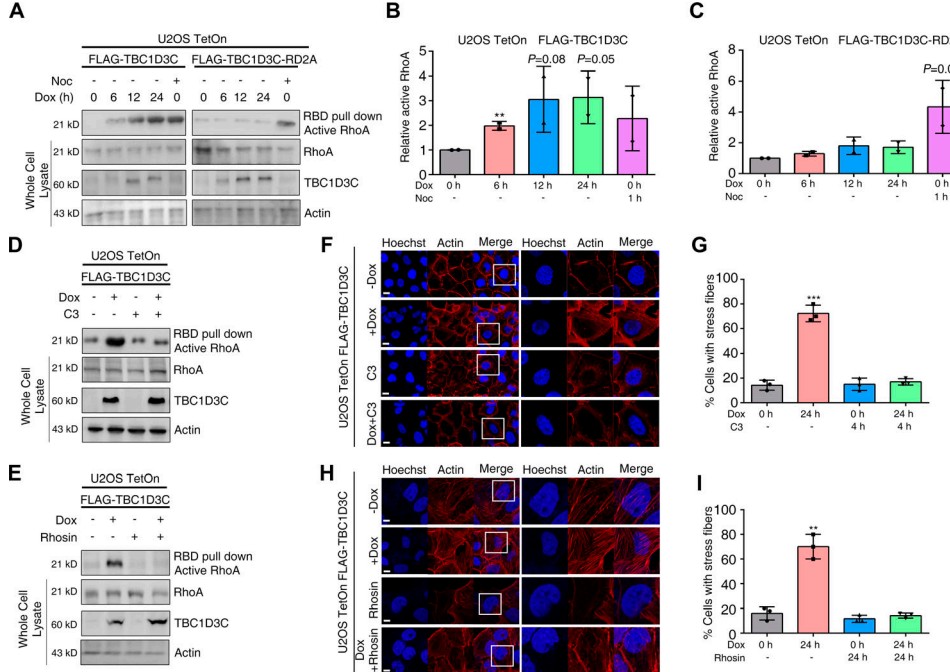

**Figure 2. RhoA acts as the mediator of TBC1D3C-induced actin filament assembly.** **(A, B, C)** TBC1D3C promotes the activation of RhoA. **(B, C)** Relative active RhoA in (A). Active RhoA was pulled down from lysates expressing TBC1D3C or TBC1D3C-RD2A using a GST-RBD fusion protein. Noc treatment was used as controls. **(D, E)** RhoA activation induced by TBC1D3C is blocked by the inhibitors C3 (1.5 $\mu g/ml$, 4 h) and Rhosin (30 $\mu M$, 24 h). Active RhoA was pulled down using a GST-RBD fusion protein from cell lysates as indicated. **(F, G, H, I)** Immunofluorescence staining shows actin stress fiber formation induced by TBC1D3C, which is abolished by C3 or Rhosin. Right panels are higher magnification views of the boxes depicted in left panels, *$P < 0.05$, **$P < 0.01$, ***$P < 0.001$ versus Cntrl group; scale bar, 20 $\mu m$.

toxin that binds to the barbed end of actin filaments inhibiting both the association and dissociation of actin subunits (De Caroli et al, 2021). We further confirmed that TBC1D3C expression led to the accumulation of polymerized filamentous actin (F-actin) and a corresponding decrease in levels of monomeric globular actin (G-actin) through centrifugation-based separation (Fig 1D–F). Collectively, these data suggest that TBC1D3C plays a role in inducing actin polymerization, shedding light on its previously unknown biological function.

## RhoA mediates actin filament assembly induced by TBC1D3C

To investigate the mechanism underlying TBC1D3C-induced actin stress fiber formation, we sought to identify the Rho GTPase that is activated by TBC1D3C. Rho family GTPases, including RhoA, RhoB, and RhoC, are known to trigger the formation of contractile stress fibers and focal adhesion complexes. To this end, we employed an affinity precipitation assay to capture activated Rho GTPases and identified candidates using mass spectrometry (see Fig S1A). We used the GST-rhotekin-Rho binding domain (GST-RBD) as a bait to capture activated Rho GTPases from lysates of control and TBC1D3C-expressing U2OS cells. Rhotekin is a protein with an N-terminal Rho GTPase binding domain that can bind to and inhibit the GTPase activity of the GTP-bound form of RhoA, RhoB, and RhoC (Reid et al, 1996).

The captured proteins were separated by SDS–PAGE and stained with Coomassie blue dye. Bands corresponding to Rho GTPase molecular weight ranges were excised and subjected to mass spectrometry analysis. The specific peptides of RhoA and quantification of its featured peptides and ion pairs revealed a significant increase in RhoA accumulation in samples expressing TBC1D3C (Fig S1A–C).

To further validate RhoA activation in TBC1D3C-expressed cells, we employed GST-RBD for the affinity precipitation of active RhoA from cell lysates, followed by detection of RhoA using Western blotting. The precipitated RhoA increased after induction of TBC1D3C, and treatment with nocodazole (Noc), a microtubule depolymerization agent, also induced RhoA activation (Fig 2A–C). In contrast, there was no significant change in RhoA activation in cells expressing the TBC1D3C-RD2A mutant (Fig 2A–C). In addition, treatment with C3 recombinant protein using the LFn-PA system reversed TBC1D3C-mediated RhoA activation, leading to a decrease in stress fiber formation (Fig 2D, F, and G). Similarly, administration of Rhosin, a RhoA inhibitor, produced a similar effect (Fig 2E, H, and I). These results indicate that RhoA may play a critical role in mediating TBC1D3C-induced actin polymerization.

## GEF-H1 mediates RhoA activation stimulated by TBC1D3C

Next, we sought to identify the GTPase exchange factors responsible for RhoA activation stimulated by TBC1D3C. To accomplish this, we incubated lysates from TBC1D3C-expressing or control U2OS cells with GST-RhoA-G17A–coupled beads and identified differentially precipitated proteins using mass spectrometry. RhoA-G17A is a RhoA mutant that binds to activated GEFs with high affinity. Among the differentially precipitated proteins, GEF-H1 showed significant enrichment in TBC1D3C-expressing cells compared with the control (see Fig S2A–E). To verify the activation of GEF-H1 in TBC1D3C-expressing cells, we used a GST-RhoA-G17A pulldown assay and found a progressive increase in GEF-H1 precipitation corresponding to elevated TBC1D3C expression upon Dox induction over time, indicating enhanced activation of GEF-H1 (Fig 3A–C). In contrast, in cells expressing the TBC1D3C mutant it did not show GEF-H1 precipitation (Fig 3A–C). In addition, TBC1D3C induced alterations in GEF-H1 localization and

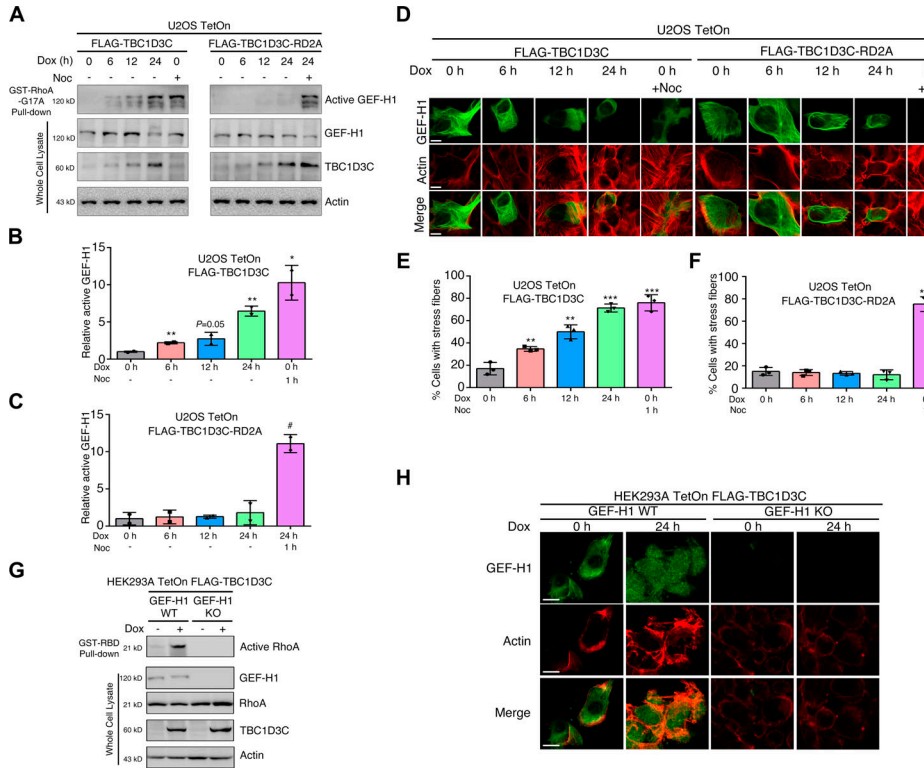

**Figure 3. GEF-H1 is a TBC1D3C-activated RhoA exchange factor.**

**(A, B, C)** TBC1D3C promotes the activation of GEF-H1. **(B, C)** Relative active GEF-H1 in (A). Active GEF-H1 was pulled down using GST-RhoA (G17A) fusion protein from lysates expressing TBC1D3C or TBC1D3C-RD2A. **(D, E, F)** Immunofluorescence staining reveals changes in actin dynamics and GEF-H1 subcellular localization in response to TBC1D3C induction or Noc (1 μM, 1 h) treatment; scale bar, 20 μm. **(E, F)** Quantification of F-actin in (D). *TBC1D3C* and *TBC1D3C*-RD2A U2OS TetOn cells with or without Dox or Noc treatment were transfected with *GFP-GEF-H1* and subjected to immunofluorescence staining using Phalloidin-iFluor 555 Reagent. **(G)** Activation of RhoA induced by TBC1D3C is abolished in *GEF-H1* knockout (KO) HEK293A cells. HEK293A TetOn *TBC1D3C* cells with or without *GEF-H1* depletion were pulled down using GST-RBD fusion protein, and immunoprecipitated and whole-cell lysates were resolved by Western blotting. **(H)** TBC1D3C induces actin filament assembly, which is abolished by GEF-H1 depletion. HEK293A TetOn *TBC1D3C* cells with or without *GEF-H1* depletion were induced with or without Dox and subjected to immunofluorescence staining using Phalloidin-iFluor 555 Reagent; scale bar, 20 μm. *$P < 0.05$, **$P < 0.01$, ***$P < 0.001$ versus Cntrl group; #$P < 0.05$, ##$P < 0.01$, ###$P < 0.001$ versus Dox, 24 h; scale bar, 10 μm.

promoted actin assembly, whereas its mutant did not (Fig 3D–F). These results suggest that TBC1D3C activates GEF-H1.

We then investigated whether GEF-H1 mediated TBC1D3C-induced RhoA activation using the CRISPR/Cas9 system to stably knockout *GEF-H1* in HEK293A TetOn *TBC1D3C* cells (Fig 3G and H). GEF-H1 ablation resulted in the abrogation of TBC1D3C-induced RhoA activation, whereas control cells exhibited normal RhoA activation (Fig 3G). In addition, GEF-H1 ablation led to reduced stress fiber formation upon TBC1D3C induction compared with control cells (Fig 3H). Our data suggest that TBC1D3C induces activation of GEF-H1, which in turn mediates RhoA activation and subsequent actin polymerization.

## TBC1D3C induces decoupling of actin filament assembly and microtubule disassembly via relocating GEF-H1 to the cytosol

After characterizing the effects of TBC1D3C on actin filament assembly, we next investigated its impact on microtubule assembly. The induction of either TBC1D3C or TBC1D3C-RD2A over time had minimal effects on the assembly of microtubules (Fig 4A). Interestingly, even though TBC1D3C expression caused no alteration in microtubule assembly, it still promoted actin polymerization, as previously shown (Fig 4B). To assess the impact of TBC1D3C on GEF-H1 localization, we examined its distribution in response to TBC1D3C expression. TBC1D3C induced GEF-H1 release from microtubules, leading to its even distribution throughout the cytoplasm, again without causing microtubule disassembly (Fig 4C–E). Intriguingly, treatment with paclitaxel

(PTX), a microtubule stabilizer, did not prevent GEF-H1 from relocation to the cytosol and actin fiber formation upon TBC1D3C expression (Fig 4B–E). Noc was used as control treatments to assess the effect of TBC1D3C on microtubule dynamics.

The dynein motor complex, consisting of Tctex-1, dynein intermediate chain (DIC), and 14-3-3, anchors GEF-H1 to microtubules (Meiri et al, 2014). To investigate the effect of TBC1D3C on the interaction between GEF-H1 and dynein motor complex, we performed immunoprecipitation experiments. GEF-H1 co-precipitated with Tctex1, DIC, and 14-3-3 (Fig 4F). However, upon TBC1D3C induction, the amount of Tctex1, DIC, and 14-3-3 present in GEF-H1 immune complexes was significantly reduced, indicating that TBC1D3C disrupts the interaction between GEF-H1, Tctex-1, DIC, and 14-3-3 (Fig 4F). In contrast, TBC1D3C-RD2A did not affect the DIC, Tctex1, 14-3-3, and GEF-H1 immunocomplex (Fig 4F), suggesting that the observed effects were specific to wild-type TBC1D3C. Together, these results demonstrate that TBC1D3C induces GEF-H1 release from microtubules and actin filament assembly independent of microtubule disassembly. In addition, we also demonstrate that TBC1D3C disrupts the interaction between GEF-H1, Tctex-1, DIC, and 14-3-3, providing a potential mechanism for TBC1D3C-mediated regulation of GEF-H1 release from microtubules.

## TBC1D3C interacts directly with GEF-H1

To further explore the mechanism by which TBC1D3C activates GEF-H1, we performed pulldown assays to test their direct

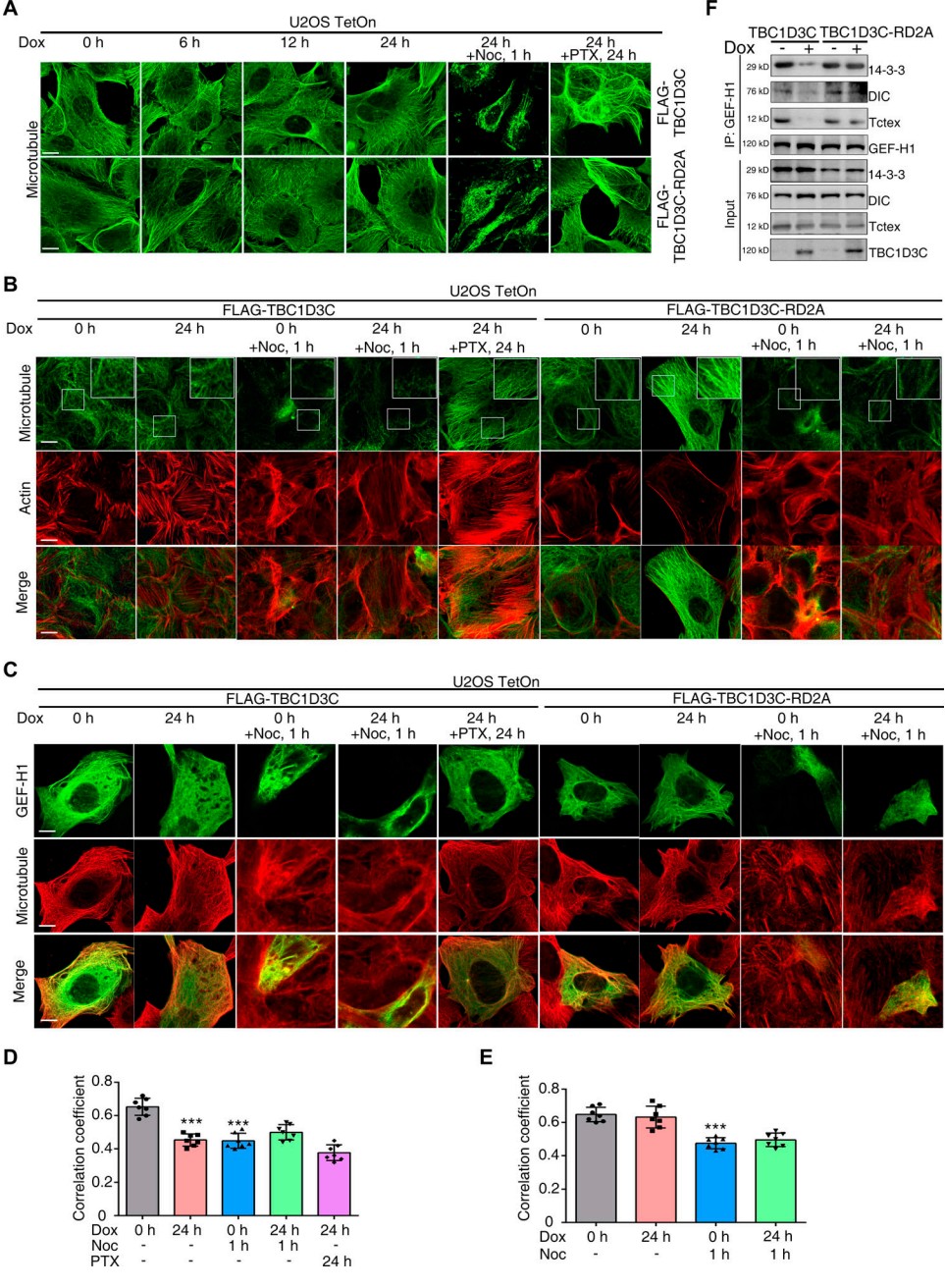

**Figure 4. TBC1D3C induces the decoupling of actin filament assembly and microtubule disassembly via relocating GEF-H1 to the cytosol.**
**(A)** Microtubule dynamics are not affected by induction of TBC1D3C or TBC1D3C-RD2A over time. Cells were treated with Dox or together with Noc (1 $\mu$M, 1 h) or PTX (200 nM, 24 h) as indicated and stained with $\alpha$-tubulin antibody; scale bar, 20 $\mu$m. **(B)** TBC1D3C promotes actin stress fiber formation without altering microtubule structure. Cells were treated as indicated and stained with $\alpha$-tubulin antibody for microtubules and Phalloidin-iFluor 555 Reagent for actin; scale bar, 20 $\mu$m. **(C, D, E)** TBC1D3C induces GEF-H1 release from the microtubule without changes in microtubule dynamics; scale bar, 20 $\mu$m. **(D, E)** Correlation coefficients of GEF-H1 and microtubules in (C). Cells were treated as indicated with *GFP-GEF-H1* transfection for GEF-H1 labeling, and stained with $\alpha$-tubulin antibody for microtubules. **(F)** TBC1D3C inhibits the binding of GEF-H1 to Tctex, DIC, and 14-3-3 protein complexes. Cells treated as indicated were subjected to immunoprecipitation with GEF-H1 antibody, and immunoprecipitated and whole-cell lysate proteins were immunoblotted with respective antibodies. Visualization of actin, microtubules, and GEF-H1 was performed by confocal microscopy. *$P$ < 0.05, **$P$ < 0.01, ***$P$ < 0.001 versus Cntrl group.

interaction. We purified MBP-GEF-H1 and GST-TBC1D3C from *E. coli*, respectively. GST-TBC1D3C efficiently pulled down GEF-H1, indicating a direct binding between TBC1D3C and GEF-H1 (Fig 5A). To identify the specific binding domain within TBC1D3C, we co-transfected Myc-tagged GEF-H1 with either the full-length (FL) TBC1D3C or its various truncation and internal deletion mutants. Among the mutants tested, the N-terminal region (residues 1–353) of TBC1D3C, which encompasses the complete TBC domain, directly interacted with GEF-H1, whereas internal deletion within this domain abolished the binding (Fig 5B and C). We also examined the importance of the conserved amino acid residues

Arg107 and Asp148 of TBC1D3C in binding with GEF-H1 using the R107A, D148A, and RD2A mutants. The R107A and D148A mutations reduced the binding, and the RD2A mutation further weakened the interaction (Fig 5D). In addition, to identify the binding domain in GEF-H1, we generated a series of Myc-tagged constructs representing its different functional domains and co-expressed them with FLAG-tagged TBC1D3C (Fig 5E). Only the Dbl-homology (DH) and pleckstrin-homology (PH) domain of GEF-H1 showed significant interaction with TBC1D3C, whereas other domains or single PH or DH domains showed minimal interaction (Fig 5F). These results suggest that the N-terminal

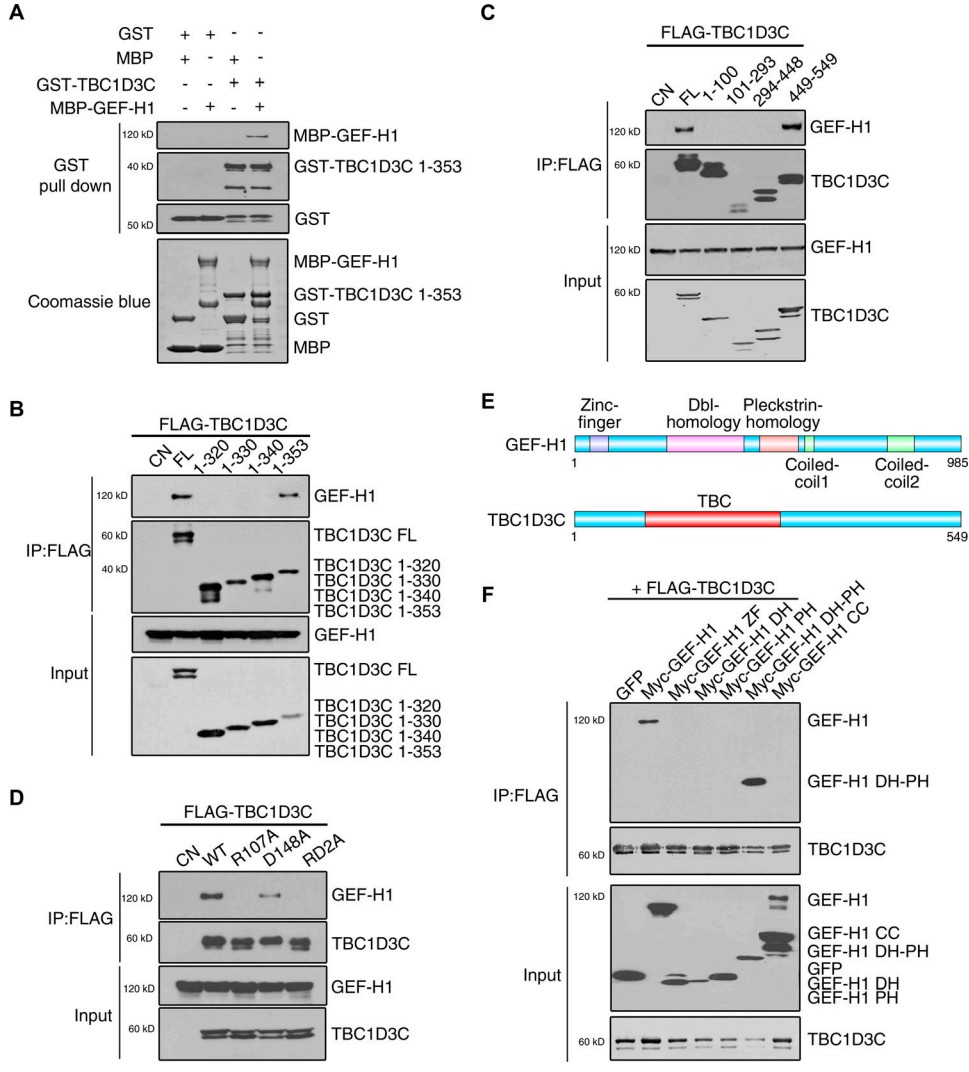

**Figure 5. Direct interaction between TBC1D3C and GEF-H1.**
**(A)** TBC1D3C interacts with GEF-H1 directly. GST or GST-TBC1D3C on glutathione Sepharose beads was incubated with MBP or MBP-GEF-H1 for pulldown assay. Pulled-down proteins were immunoblotted with MBP or GST antibodies. **(B, C)** TBC domain (residues 1–353) of TBC1D3C mediates its GEF-H1 interaction. HEK293T cells were co-transfected with full-length (FL) *TBC1D3C* or its truncation (B) or internal deletion mutants (C) and *Myc-GEF-H1*. Immunoprecipitation was done with anti-FLAG antibody–conjugated agarose M2 beads, followed by immunoblotting with FLAG and Myc antibodies. **(D)** Residues R107 and D148 of TBC1D3C mediate its interaction with GEF-H1. FLAG-TBC1D3C wide type (WT) or its mutants R107A, D148A, or RD2A were co-expressed with Myc-GEF-H1 in HEK293T cells followed by immunoprecipitation and immunoblotting as indicated. **(E, F)** Dbl-homology (DH) and pleckstrin-homology (PH) domains of GEF-H1 interact with TBC1D3C. **(E)** Schematic of GEF-H1 and TBC1D3C domain structures. **(F)** FLAG-TBC1D3C was co-expressed with Myc-GEF-H1 or its deletion mutants in HEK293T cells, followed by immunoprecipitation and immunoblotting as indicated.

TBC domain of TBC1D3C directly interacts with the DH-PH domain of GEF-H1.

## Dynamics of TBC1D3C-induced decoupling of actin filament assembly and microtubule disassembly

To obtain super-resolution images of cytoskeleton changes, we labeled actin, microtubule, and GEF-H1 in U2OS cells and imaged them using structured illumination microscopy (SIM). GEF-H1 exhibited co-localization with microtubules, but became uniformly distributed throughout the cytoplasm after TBC1D3C induction by Dox treatment (Fig 6A and B). Concurrently, more actin stress fibers were formed (Fig 6A and C). Notably, microtubule organization showed minimal changes in response to TBC1D3C expression (Fig 6A). For live-cell imaging of GEF-H1, microtubules, and actin filament dynamics, we labeled actin with SiR-actin and transfected cells with *GFP-GEF-H1* and *mCherry–tubulin* plasmids. Using a DeltaVision live-imaging system, upon the induction of TBC1D3C expression over time, we observed a gradual increase in bundled actin fibers (red) and a progressive release of GEF-H1 (green) from microtubules, whereas microtubule (magenta) organization remained largely unchanged (Fig 6D and E and Video 1, Video 2, Video 3, and Video 4). To further verify the dissociation of GEF-H1 from microtubules, centrifugation-based analysis of its distribution was performed. In cells expressing TBC1D3C, we observed an increased accumulation of GEF-H1 in the supernatant and a decreased level of GEF-H1 in the pellet, indicative of its release from microtubules (Fig 6F). In contrast, cells expressing the TBC1D3C-RD2A mutant showed no significant alterations in GEF-H1 distribution (Fig 6F). These findings suggest that TBC1D3C plays a role in decoupling actin filament assembly from microtubule disassembly by modulating the subcellular localization of GEF-H1.

## Discussion

In this study, we investigated the biological function of TBC1D3C and unveiled its regulatory role in actin and microtubule dynamics. Our

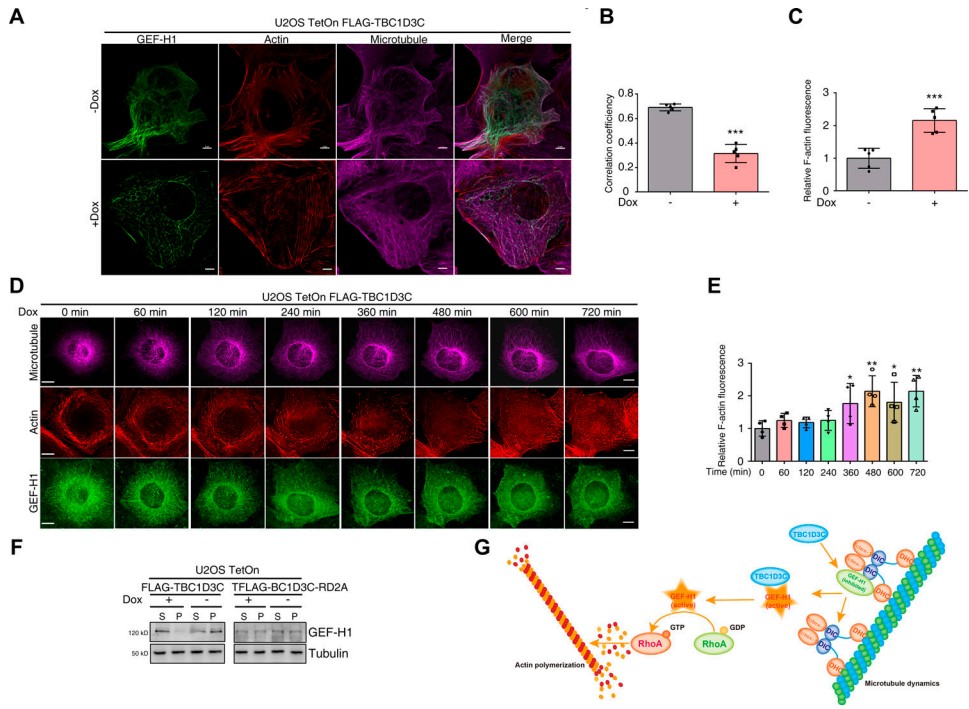

**Figure 6.  Dynamics of TBC1D3C-induced decoupling of actin filament assembly and microtubule disassembly.**
**(A, B, C)** Structured illumination microscopy imaging showing GEF-H1 relocalization from microtubules into the cytosol and actin stress fiber formation without microtubule disassembly; scale bar, 5 µm. **(B)** Correlation coefficients of GEF-H1 and microtubules in (A). **(C)** Quantification of F-actin in (A). *TBC1D3C* U2OS TetOn cells with or without Dox treatment were transfected with *GFP-GEF-H1*, fixed, and stained with phalloidin and anti-tubulin antibodies. Images were captured by structured illumination microscopy. **(D, E)** Time-lapse live-cell imaging of microtubule, actin, and GEF-H1 dynamics after TBC1D3C induction. **(E)** Quantification of F-actin in (D). Cells were labeled with SiR-actin for actin (red) and transfected with *GFP-GEF-H1* (green) and *mCherry–tubulin* (magenta); scale bar, 20 µm. **(F)** TBC1D3C expression leads to GEF-H1 accumulation in the cytoplasm. Lysates from TBC1D3C- or TBC1D3C-RD2A–expressing cells were subjected to centrifugation. GEF-H1 and microtubules in the supernatant and pellet were determined by Western blotting with respective antibodies. **(G)** Proposed model for TBC1D3C-directed GEF-H1 activation and decoupling of actin filament assembly and microtubule disassembly. GEF-H1 is tethered to polymerized microtubules by the dynein motor light-chain Tctex-1. TBC1D3C interacts directly with GEF-H1 to promote its release from microtubules to the cytosol without microtubule disassembly, leading to RhoA activation and subsequent actin filament assembly. *P < 0.05, **P < 0.01, ***P < 0.001 versus Cntrl group.

findings suggested that TBC1D3C functions as a GEF activator to induce GEF-H1 release from microtubules without altering microtubule organization. Release of GEF-H1 into cytosol leads to RhoA activation and subsequent actin polymerization (Fig 6G). Our findings demonstrated that through this mechanism, TBC1D3C induces the decoupling of actin assembly and microtubule disassembly, a unique activity of TBC1D3C, shedding light on a novel regulatory mechanism in cellular cytoskeleton dynamics. Although we observed the decoupling effect of TBC1D3C, the physiological and pathological significance of this decoupling of actin assembly and microtubule disassembly requires further investigation.

Previous studies have reported similar findings in the context of G protein–coupled receptor (GPCR) signaling. Ligands such as LPA or thrombin can stimulate the disassociation of the GEF-H1:dynein multiprotein complex independent of microtubule depolymerization (Meiri et al, 2014), which results in the release of GEF-H1, thereby promoting RhoA activation and actin polymerization. In addition, GEF-H1 can be activated during interphase through a distinct mechanism that is independent of microtubule disassembly (Guilluy et al, 2011). In our study, we identified a novel activation paradigm for GEF-H1 that is independent of microtubule disassembly, involving direct protein interaction between GEF-H1 and TBC1D3C. We also characterized the binding domains responsible for the interaction between TBC1D3C and GEF-H1 and identified that the N-terminal TBC domain of TBC1D3C and the PH-DH domain of GEF-H1 are responsible for their mutual interaction.

Several questions remain to be addressed regarding TBC1D3C and its regulatory role in GEF-H1. Firstly, the upstream regulators of TBC1D3C are currently unclear. A yeast two-hybrid screening of TBC1D3C against a human cDNA library yielded potential binding targets for TBC1D3C, including nischarin, an anti-apoptotic protein that regulates Rac-1–dependent cell motility. Nischarin preferentially binds to the cytoplasmic domain of the integrin α5 subunit, inhibiting cell motility and altering actin filament organization (Alahari et al, 2000). The fibronectin receptors α5β1 integrin and syndecan-4 co-cluster in focal adhesions and orchestrate cell migration, contributing to the suppression of RhoA activity during matrix engagement (Bass et al, 2008). We hypothesize that external stimuli promote nischarin release mediated by α5β1 integrin, leading to activation of TBC1D3C.

TBC1D3C contains a TBC domain initially reported to be a Rab5-GAP (Frittoli et al, 2008). However, the TBC domain within TBC1D3C lacks its critical arginine residue and is replaced by a glycine, and therefore, no strong enzymatic activity was found in TBC1D3C (Martinu et al, 2002; Haas et al, 2005; Bizimungu et al, 2007). We demonstrated its interaction with GEF-H1 to stimulate GEF-H1 release from microtubules for its activation. Our study uncovers probably one of the biological functions of TBC1D3C.

GEF-H1 harbors a cysteine-rich zinc finger–like motif at its amino terminus and a coiled-coil domain at its carboxy terminus. GEF-H1 exhibits guanine nucleotide exchange activity for RhoA but not for Rac1 or Cdc42 (Brecht et al, 2005). Its activity is suppressed by the interaction of its C-terminus with microtubules and by binding to tubulin or other microtubule-associated proteins (Zenke et al, 2004). Microtubule depolymerization leads to GEF-H1 release into the cytosol for activation, followed by RhoA activation and assembly of the actin cytoskeleton (Krendel et al, 2002). Thus, GEF-H1 mediates the coupling of actin filament polymerization and microtubule depolymerization. Interestingly, in our study, TBC1D3C

directly binds to GEF-H1, which induces GEF-H1 relocation to the cytoplasm from microtubules without microtubule depolymerization, leading to its activation and subsequent RhoA activation and actin filament polymerization. Our study unveils a new activation mode of GEF-H1 and broadens our understanding of its regulation in cytoskeleton organization.

In conclusion, our findings reveal a novel mechanism whereby actin filament assembly is decoupled from microtubule disassembly, providing insights into the intricate regulation of the cytoskeleton. Moreover, our findings shed light on the function of a recently evolved primate-specific gene.

# Materials and Methods

### Cell lines and reagents

HEK293T was kindly provided by Dr. Long Zhang (Zhejiang University, China). HEK293A was kindly provided by Dr. Bin Zhao (Zhejiang University, China). The U2OS cell was provided by Dr Fangwei Wang (Zhejiang University, China). Cells were grown in DMEM (Invitrogen) supplemented with 10% heat-inactivated FBS, 100 U/ml of penicillin, and 100 mg/ml of streptomycin at 37°C in 5% $CO_2$.

Rabbit antibodies against $\beta$-actin, GAPDH, and Myc were purchased from HuaBio-Antibodies; rabbit antibodies against RhoA and GEF-H1 were purchased from Cell Signaling Technology; mouse antibodies against TBC1D3C were purchased from Santa Cruz Technology; antibodies against Tctex, DIC, and 14-3-3 were purchased from Proteintech; mouse antibody against $\alpha$-tubulin was purchased from Life Technologies; goat anti-rabbit and goat anti-mouse IgG-conjugated horseradish peroxidase antibodies were purchased from Promega; and Alexa Fluor–conjugated goat anti-mouse and rabbit IgG antibodies were bought from Jackson ImmunoResearch.

M2-conjugated magnetic beads were purchased from Sigma-Aldrich. Nickel-agarose beads and immobilized glutathione were purchased from Thermo Fisher Scientific. Amylose resin was purchased from New England Biolabs.

Rhosin was bought from MCE.

### Plasmids

pLX-sgRNA was purchased from Addgene. pCMV5-TetOn *TBC1D3C* and pCMV5-TetOn *TBC1D3C* (R107A, D148A, RD2A) were generated by subcloning a fragment of the human *TBC1D3C* cDNA and inserted into pCMV5-TetOn vectors, respectively. Deletion mutants *TBC1D3C* 1–353, *TBC1D3C* 1–320, *TBC1D3C* 1–330, *TBC1D3C* 1–340, *TBC1D3C* Δ1-100, *TBC1D3C* Δ294-448, *TBC1D3C* Δ449-549, *GEF-H1* ZF, *GEF-H1* DH, *GEF-H1* PH, *GEF-H1* DH-PH, and *GEF-H1* CC were obtained by PCR, respectively. *TBC1D3C* R107A and D148A mutants were generated by site-directed mutagenesis PCR (Quick Change Mutagenesis Kit; Stratagene).

### Cell transfection

Plasmid DNA transfection of HEK293T and HEK293A cells was performed using polyethylenimine. Transfection in U2OS cells was performed by Lipofectamine 3000.

*GEF-H1*–deficient HEK293T cells were generated using CRISPR/Cas9-mediated gene editing. Two guide RNAs (GCACATGGTCATGCCGGAGA and GACAAGGTAGGAGTCAGCCT) targeting *GEF-H1* were designed, synthesized by GenePharma, and cloned into the pSpCas9(BB)-2A-GFP (PX458) vector (plasmid #48138; Addgene). After transfection, HEK293T cells were sorted according to GFP expression, expanded, and subsequently screened for GEF-H1 expression by Western blot.

### Stable cell lines

HEK293A and U2OS cells were transfected with TetOn-*TBC1D3C* or TetOn-*TBC1D3C*-RD2A mutant using Lipofectamine 3000. Four hours after transfection, media were removed, and fresh media were added. 24 h after infection, the media were changed to fresh media containing 1 μg/ml puromycin. Puromycin-resistant cell pools were used for further experiments.

### Immunofluorescence

Cells were fixed in 4% PFA for 15 min and then washed with PBS three times. After that, the cells were permeabilized with PBS containing 0.1% Triton X-100 for about 15 min and incubated in PBS containing 0.1% Triton X-100 and 5% BSA for 1 h. Subsequently, the cells were incubated sequentially with Phalloidin-iFluor 555 Reagent (ab176756; Abcam), indicated primary antibodies, and Alexa Fluor–labeled secondary antibodies with extensive washing. DAPI at a concentration of 0.5 μg/ml in PBS containing 0.1% Triton X-100 was used to stain the nuclei. Immunofluorescence images were obtained with a Zeiss confocal microscope.

### Quantitative image analysis

Stress fiber formation was quantified through manual inspection, where cells exhibiting lateral stress fibers were counted. A stress fiber was defined as an actin filament spanning the lateral width of the cell; if a cell displayed one or more stress fibers, it was classified as positive, and if not, as negative.

To determine the correlation coefficients, ImageJ was used for image processing, which offers several pre-installed plugins, including a co-localization analysis procedure. Specifically, Pearson's correlation coefficient was employed to assess co-localization parameters in our study. This approach allows for robust analysis of the spatial relationship between the analyzed images.

### Actin segmentation by centrifugation

Cells were homogenized with RIPA lysate (#P0013B; Beyotime) with 1% phenylmethanesulfonyl fluoride (#ST506; Beyotime) for 30 min on ice before being collected. Cell lysates were subjected to centrifugation at 15,000$g$ for 30 min at 4°C. The supernatant containing G-actin was transferred to a fresh tube, whereas the pellet containing F-actin was resuspended in cold PBS and then centrifuged at 15,000$g$ for 5 min twice. After centrifugation, the pellet was resuspended in F-actin extracting solution (1.5 mM guanidine hydrochloride, 1 mM sodium acetate, 1 mM $CaCl_2$, 1 mM ATP, 20 mM Tris–HCl, pH 7.5) for 1 h on ice to dissolve F-actin before being centrifuged at 15,000$g$ for 30 min at 4°C. Then, F-actin and G-actin

became monomers, which were analyzed using an antibody against β-actin by immunoblotting.

### Microtubule co-sedimentation

Cells were washed twice with PBS and then extracted by microtubule-stabilizing buffer (100 mM Pipes, pH 6.9, 5 mM MgCl$_2$, 2 mM EGTA, 2 M glycerol, 0.1% NP-40, 10 mM β-glycerophosphate, 50 mM NaF, 0.3 mM okadaic acid (Roche), 1 mM PMSF, and protease inhibitor cocktail) for 15 min at RT. This extract was separated at 1,000$g$ for 5 min at RT, which is sufficient to pellet both the unextractable and the microtubule-enriched cytoskeleton components. The soluble cytosolic supernatant contains major parts of the total cell protein including unassembled tubulin.

### Immunoblotting

Cells were treated as indicated in the figure legends and washed with cold phosphate-buffered saline. Lysis buffer (20 mM Tris–HCl, pH 7.4, 150 mM NaCl, 0.5% NP-40, 10% glycerol, 1 mM DTT, and cOmplete protease inhibitor cocktail) was added for 10 min on ice and centrifuged at 10,000$g$ for 10 min. The protein concentration was measured, and equal amounts of proteins were then separated by SDS–PAGE, transferred to PVDF membranes (Merck Millipore), blocked, and detected with antibodies specified in figures.

### Protein purification

The lysis buffer for different tagged proteins is different owing to the beads. According to the manufacturer's instructions, His-tagged and GST-tagged proteins were purified using Nickel-agarose and glutathione Sepharose beads. According to the manufacturer's instructions, MBP-tagged proteins were purified using amylose resin. FLAG-tagged proteins were purified using anti-FLAG M2 magnetic beads. Briefly, proteins were expressed in the BL21(DE3) pLysS strain of *Escherichia coli*. After expression induction with 0.1 mM isopropyl β-D-thiogalactoside at 16°C overnight, the bacteria were centrifuged and resuspended in cold lysis buffer (50 mM Tris–HCl, pH 7.6, 50 mM NaCl, 5 mM MgCl2, 0.5% NP-40, 1 mM DTT, 1 mM PMSF, and protease inhibitors) for sonication and centrifugation, and then, the supernatant was added with pre-equilibrated glutathione Sepharose beads (GE Healthcare) and rotated at 4°C for 2 h. After centrifugation, the beads were washed twice in lysis buffer. Finally, bound proteins were resuspended gently in the wash buffer containing 1 mM DTT, 1 mM PMSF, protease inhibitor cocktail, and 10% glycerol and were stored at 4°C for subsequent use.

### GST pulldown assay, immunoprecipitation, and protein identification

HEK293A cells were lysed on ice with lysis buffer (50 mM Tris–HCl, pH 7.5, 100 mM NaCl, 1% Triton X-100, 10 mM NaF, 1 mM Na3VO4, 1 mM EDTA, 1 mM EGTA, 1 mM PMSF, and protease inhibitor cocktail), followed by centrifugation at 17,949$g$ for 20 min. For GST pulldown assays, the supernatants were incubated with GST-RBD or GST-RhoA (G17A) beads, and the binding mix was incubated for 3 h at 4°C. After washing with wash buffer (50 mM Tris–HCl, pH 7.6,

50 mM NaCl, 5 mM MgCl$_2$, and 1 mM DTT) for five times, the proteins associated with beads were boiled and subjected to electrophoresis.

The SDS–PAGE were stained with Coomassie staining, and the protein bands were excised and minced into small pieces. The samples were reduced with 1 mM dithiothreitol for 30 min at 60°C and then alkylated with 5 mM iodoacetamide for 45 min in the dark at RT. Then, the gels were digested with trypsin. Gel pieces were washed and dehydrated with acetonitrile for 10 min, followed by removal of acetonitrile. Peptides were washed with a solution containing 50% acetonitrile and 5% acetic acid. The extracts were dried in a Speed-Vac apparatus (1 h).

The samples were stored at 4°C until analysis. The samples were reconstituted in high-performance liquid chromatography (HPLC) solvent A (2.5% acetonitrile and 0.1% formic acid) and analyzed with the nanoflow capillary liquid chromatography–tandem mass spectrometry (LC/MS/MS) system using a capillary LC system (Thermo Fisher Scientific) coupled to a quadrupole time-of-flight (Q-TOF) mass spectrometer. Eluting peptides were detected, isolated, and fragmented to produce a tandem mass spectrum of specific fragment ions for each peptide (Yang et al, 2022). Peptide product ion spectra produced by LC/MS/MS were screened against the SwissProt protein database using the Mascot search program sequence database searching engine sequences.

For the PRM analysis, we selected unique peptides for our proteins of interest. The corresponding peptide information, including their retention time, m/z value, and ion charge, was obtained from data-dependent acquisition data. This precursor ion information was exported to an Excel file and used for PRM analysis. PRM samples were prepared similar to data-dependent acquisition samples, with each group having four replicates. During PRM acquisition, the peptides of interest were selectively fragmented, and both their precursor and fragment ions were detected. A retention time window of ±10 min was used to ensure peptide detection, and fewer than 30 peptides were analyzed simultaneously to maintain a reasonable cycle time. The acquired PRM data were directly imported into Skyline to quantify the precursor ions with their three isotopic peaks and the top five fragment ions (b ions and y ions). The related peptide information was then exported, and the peak areas of the fragment ions were summed up as the peptide signal intensity. Finally, the selected peptides were integrated as protein levels.

### Rho GTPase pulldown assay

Rho activation detection was performed as previously reported. Briefly, cells treated as indicated were starved for 3 h and lysed with ice-cold buffer (20 mM Hepes, pH 7.4, 0.1 M NaCl, 1% Triton X-100, 10 mM EGTA, 40 mM β-glycerophosphate, 20 mM MgCl$_2$, 1 mM Na$_3$VO$_4$, 1 mM dithiothreitol, 10 µg/ml aprotinin, 10 µg/ml leupeptin, and 1 mM phenylmethylsulfonyl fluoride). The lysates were mixed with GST-RBD previously bound to glutathione Sepharose beads and incubated for 2 h at 4°C. After washing with lysis buffer three times, the pulldown Rho was detected by SDS–PAGE with RhoA primary antibody.

### Detection of activated GEF-H1

Purified GST-tagged RhoA G17A has a high affinity toward active GEF, allowing it to pull down active GEF, followed by Western blotting using a GEF-H1-specific antibody to detect the presence of the endogenous or GEF-H1 proteins, respectively. The experimental procedure is as same as the RBD pulldown assay. GEF-H1 in total cell lysates was also detected for each sample.

### Immunoprecipitation

HEK293T cells transfected with indicated plasmids were lysed and subjected to immunoprecipitation for immunoprecipitation studies. Cell lysates were immunoprecipitated with anti-FLAG antibody–conjugated agarose M2 beads. Then, the immunoprecipitated proteins were resolved by 2 × SDS loading buffer, analyzed by SDS–PAGE, and immunoblotted with indicated antibodies.

### Imaging of fixed cells with SIM

U2OS cells were observed with a 100× oil objective lens with a numerical aperture of 1.49 on a Nikon N-SIM (Nikon Corporation). Phalloidin-iFluor 488 Reagent was excited with the 488-nm wavelength. Alexa Fluor 555–labeled probes were excited with the 555-nm wavelength.

### Live-cell imaging

For microtubule and actin dynamic observation, U2OS cells stably transfected with TetOn-*TBC1D3C* were treated with Dox and imaged with a GE Healthcare DeltaVision Ultra high-resolution microscope (DeltaVision OMX SR) at 37°C and 5% $CO_2$ with a 60× oil objective. Time-lapse sequences of fluorescent images were taken for 16 h (after Dox induction) at 30-min intervals, followed by image deconvolution and maximum intensity quick projection.

### Statistical analysis

A two-tailed unpaired *t* test was used to conduct statistical analysis between two groups. Differences in means were considered significant if $P < 0.05$.

## Data Availability

Further information on materials, datasets, and protocols should be directed to zxia2018@zzu.edu.cn.

## Supplementary Information

## Acknowledgements

This work was partially funded by the National Natural Science Foundation of China, grant nos. 31970158 and 32470159 to Z Xia.

### Author Contributions

Y Luan: conceptualization, data curation, formal analysis, and writing—original draft.
Z Deng: methodology, resources, data curation, and formal analysis.
Y Zhu: data curation and formal analysis.
L Dai: data curation and formal analysis.
Y Yang: data curation, formal analysis, and methodology.
Z Xia: conceptualization, supervision, funding acquisition, and writing—original draft, review, and editing.

### Conflict of Interest Statement

The authors declare that they have no conflict of interest.

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
