## [Reviewer comments · Life Science Alliance]

Life Science Alliance

Decoupling actin assembly from microtubule disassembly by TBC1D3C-mediated direct GEF-H1 activation

Yi Luan, Zhifeng Deng, Yutong Zhu, Lisi Dai, Yang Yang, and Zongping Xia

DOI: [10.26508/lsa.202402585](https://doi.org/10.26508/lsa.202402585)

Corresponding author(s): Zongping Xia, First Affiliated Hospital of Zhengzhou University

Review Timeline:

Submission Date:	2024-01-10
Editorial Decision:	2024-03-01
Revision Received:	2024-08-14
Editorial Decision:	2024-09-09
Revision Received:	2024-10-09
Editorial Decision:	2024-10-10
Revision Received:	2024-10-16
Accepted:	2024-10-16

Transaction Report:

March 1, 2024

Re: Life Science Alliance manuscript #LSA-2024-02585-T

Dr. Zongping Xia
First Affiliated Hospital of Zhengzhou University
1st Jianshe Road, Zhengzhou, Henan
Zhengzhou 450052
China

Dear Dr. Xia,

Thank you for submitting your manuscript entitled "Decoupling actin assembly from microtubule disassembly by TBC1D3C-mediated direct GEF-H1 activation". The manuscript has been evaluated by expert reviewers, whose reports are appended below. Unfortunately, after an assessment of the reviewer feedback, our editorial decision is against publication in Life Science Alliance.

Although your manuscript is intriguing, I feel that the points raised by the reviewers are more substantial than can be addressed in a typical revision period. If you wish to expedite publication of the current data, it may be best to pursue publication at another journal.

Given the interest in the topic, I would be open to re-submission to Life Science Alliance of a significantly revised and extended manuscript that fully addresses the reviewers' concerns and is subject to further peer review. If you would like to resubmit this work to Life Science Alliance, you may submit an appeal directly through our manuscript submission system. Please note that priority and novelty would be reassessed at re-submission.

Regardless of how you choose to proceed, we hope that the comments below will prove constructive as your work progresses.

Thank you for thinking of Life Science Alliance as an appropriate place to publish your work.

Sincerely,

Reviewer #1 (Comments to the Authors (Required)):

Luan et al. "Decoupling actin assembly from microtubule disassembly by TBC1D3C-mediated direct GEF-H1 activation".

In this manuscript, the authors present the results of experiments designed to determine the effect of TBC1D3C on the actin cytoskeleton and the underlying mechanism. TBC1D3C is a Rab-family GAP lacking the catalytic arginine, therefore has little or no GAP activity. Its expression is restricted to primates. The first present IF data supporting a function in F-actin formation and stress fiber formation. They interpreted their results to indicate a mutant of TBC1D3C did not. They then present evidence that the effect is mediated by RhoA. A proteomic approach identified GDF-H1 as a potential target. Using cells with GEF-H1 knocked out, the effects on RhoA appeared mediated by this exchange factor. The authors interpret their results examining microtubules as TBC having no effect, but that TBC caused the dissociation of GEF-H1 from microtubules. In a series of experiments using both bacterially expressed protein and cells expressed in mammalian cells, they determine that there might be direct binding of RBD to the DH-PH tandem of GDF-H1. They then tried to corroborate their results using SIM and live cell imaging. Although they present an interesting idea, the results are not sufficiently developed to be published at this time. The major problems are 1. IF that is difficult to interpret 2. Lack of replicates 3. Lack of quantification 4. Insufficient sampling of cells to draw conclusions. The shortcomings are described in more detail below.

1. Differences in actin structures are not obvious in figure 1. Furthermore, actin stress fibers can be quantified from images and should be in this case. In addition, a sufficient number of cells need to be examined, as there is usually significant variability in actin structures within a single cell population.
2. In figure 1B, only one experiment with no replicates and no quantification is shown, precluding interpretation.
3. Quantification by mass spec is very difficult and the appropriate steps to accomplish this were not used.

4. In figure 2A, the total amount of RhoA is changing. There is no normalization so can't be interpreted. In addition, a single experiment, no replicates, no quantification precluding interpretation.
5. On line 152, I think actually referring to fig 3, not fig 2 as written. Some problems as with comment 1.
6. Fig 3A is a single experiment, no replicates, no quantification, no normalization, precluding interpretation, and mutant data not clear.
7. In line 186, state no alteration in microtubules, but the conclusion is not accurate because there might be a change in modification of tubulin. It is very difficult to detect changes in microtubule morphology in cells. Dynamics also were not examined.
8. GEF-H1 distribution results have the some shortcoming described above for the other data
9. In both SIM and live cell experiments, conclusions are based on single cells. See comment 1.

Minor: the authors don't explain why they selected the particular mutants of TBC to examine.

In short, although interesting, the work is at a preliminary stage and publication is not warranted.

Reviewer #2 (Comments to the Authors (Required)):

This manuscript by Luan et al. describes the role of TBC1D3C in GEF-H1-mediated RhoA activation. The authors argue that actin and microtubule dynamics are decoupled and that TBC1D3C can have an important role in this process. The manuscript certainly has merits but the data provided does not fully support the title of the manuscript "decoupling actin from microtubule disassembly by TBC1D3C-mediated direct GEF-H1 activation". It is also not clear what the authors mean with "decoupling", does it mean that actin and microtubule is occurring simultaneously and that TBC1D3C somehow disrupts this coordination? The authors need to be more specific about what they mean. It is not clear why the authors started to study TBC1D3C. If it is only expressed in hominoids, it cannot be such a global factor in the communication between actin and microtubule assembly. What about this process in other species? Have all proteins with RabGAP domains a similar function? Do U2OS cells express TBC1D3C? The authors have created stable a cell line with inducible expression of TBC1D3C, i.e. what they study is an alteration in changes in steady state conditions over time, not a dynamic process. How is TBC1D3C activated under normal conditions? It is not clear how the observation from this lab-generated model system compares to the in vivo situation. With this said, I think the study is in most aspects well done but there are several week points that will need some input from the authors.

Detailed comments:

1. line 123 and Fig.1B. The authors describe the increase in F-actin and a corresponding done. It is not described in the Mat&Meth section. What is the buffer? Conditions for ultracentrifugation? The authors do not relate the amount of G-actin to the amount of total actin (i.e. before centrifugation). I also suggest that the authors make an effort to quantify the stress fiber formation in Fig.1A.
2. line 132. The authors claim that they have made an unbiased affinity purification. I do not agree, since there is a bias for the Rho subgroup. What about Rac and Cdc42? Alterations in their activities could also affect the status of actin.
3. Line 162, the authors found an accumulation of GEF-H1 but it is not clear if there were other RhoGEFs in the precipitate and if GEF-H1 was cherry picked because it followed the hypothesis.
4. Line 171, It is not clear what the authors mean by GEF-H1 activation? Activation in what sense? Increased binding?
5. Extended data figures does that mean supplementary figures? This is a bit confusing.
6. Fig.3B and the panels showing cells onward. Here is the problem with showing single cells. How do we know that they are representative?
7. Fig 4B. The microtubule staining is very difficult to follow in these minuscule sub panels. Would it be possible to show magnified section of some of the key conditions?
8. Fig 4C. Again, the low resolution makes it difficult to evaluate the alleged effects on GEF-H1 relocalization. I suggest that the authors perform co-localization experiments in a more quantitative manner.
9. 6A the superresolution experiments are not really convincing. I suggest that the authors use another color than blue for microtubules (in 6A and B).
10. Fig. 6CIt is not clear how the sedimentation assay has been done. Gentrification, how?
11. Fig.5E I suggest that the authors also include a figure of the domain organization of TBC1D3C. How many amino acid residues in GEF-H1 and TBC1D3C?

Reviewer #3 (Comments to the Authors (Required)):

Overview: in their work , Luan et al. present "Decoupling actin assembly from microtubule disassembly by TBC1D3C-mediated direct GEF-H1 activation" a mechanism by which GEF-H1 mediated RhoA activation can be accomplished independently of the state of the microtubule array. They propose a method by which TBC1D3C can bind GEF-H1, thereby disrupting its coupling to the Tctex-DIC-14-3-3 complex and displacing it into the cytosol where it can get activated and promote RhoA activation, which ultimately leads to the assembly of filamentous actin and stress-fiber formation.

Their work consists of measuring RhoA activity in cells expressing a doxycycline-inducible variant of TBC1D3C, where they show that induction leads to a significant increase in active RhoA in U2OS cells. The Authors show that this corresponds to an

increase in filamentous actin as measured using western blots and phalloidin stainings at different timepoints after induction of TBC1D3C expression (Figures 1 and 2). Next, Luan et al. present results that illustrate the increase of active GEFH1 in their Tet-inducible system correlated with an increase in cytosolic GEFH1 as imaged using transfected GEFH1 in immunofluorescence experiments. A knockout of GEF-H1 seems to fully abolish active RhoA and leads to less f-actin as shown using phalloidin staining (Figure 3). This seems to be independent of the state of the microtubule cytoskeleton, as GEFH1 could get released from microtubules without the microtubules getting disrupted. Additionally, the expression of TBC1D3C disrupts the interaction of GEF-H1 with the Tctex-1/DIC-DHC complex, which suggests that TBC1D3C directly binds GEFH1 and leads to its release from microtubules. A series of truncation and deletion experiments suggest that GEFH1 DH-PH domain interacts with the N-terminal domain of TBC1D3C.

Main point 1: TBC1D3C promotes F-actin filament formation. Here the authors used their TBC1D3C inducible cell line to test for an increase in F-actin formation during induction of protein expression. They provide clear results using ultracentrifugation, showcasing an increase in F-actin formation after expression induction. They also show IF images labeled with phalloidin. The way the fluorescence data is presented now might provide excerpts of individual cells; however, quantification of this is required to clearly validate this result. A very straightforward way to assess this would be to simply quantify mean intensity per cell in a series of fields of view. This simple and straightforward quantification will strengthen the result.

Main point 2: RhoA as the mediator of TBC1D3C-induced filament assembly. Here, the authors pinpoint RhoA as the mediating factor of increased filament assembly, which they again showcase using western blotting and immunofluorescence. The RBD pull-down assay of RhoA provides evidence that active RhoA does increase with increased expression time of TBC1D3C. Similarly, to the point above, quantification of fluorescence intensity would significantly strengthen the observed results. Additionally, what would also provide increased evidence is to measure active RhoA with fluorescence biosensors. Such a strong increase in RhoA activity as measured using the RBD pull-down would be clearly visible using any of the common RhoA activity reporters. However, since these could potentially be challenging experiments, I would not say that I request this experiment strictly, although in my opinion it would significantly strengthen the results.

Main Point 3: GEFH1 as the mediator between RhoA activation and TBC1D3C. Here the authors provide evidence that GEF-H1 is the connection between active RhoA and TBC1D3C. Using the GST-RhoA-G17A mutant, which binds preferentially to GEFs, after 24h of doxycycline-induced expression of TBC1D3C, they see a strong signal in the pull-down western blot of GEFH1. Additionally, using transiently expressed GEFH1 and phalloidin, they show that GEFH1 dissociates from microtubules upon induction of TBC1D3C expression, similar to what treatment with nocodazole yields. Again, quantification of the fluorescence data would provide more evidence complementing the qualitative nature of the fluorescence data currently provided. Secondly, for the knockout experiments using CRISPR-Cas, it seems that the authors used a transiently expressing cell line.

In my opinion it does not really make sense to use a transiently expressing cell line to showcase the validity of the knockout, at least not without quantifying the number of cells that show fluorescence, as the expression is transient anyway, and many cells also do not express the GEFH1 construct, which essentially renders the first panels in the knockout experiment not very informative. I would suggest that either the authors remove the top panels showing the transiently expressed GEFH1 data, provide some quantitative information about the number of cells expressing the construct, or probably best would be to repeat the experiments for a cell line expressing GEFH1 stably; otherwise, the information content is very limited since, like this, it does not provide additional information as the expression is anyway transient and thus an absence of fluorescence in single cells does not prove that the knockout was successful.

Main Point 4: TBC1D3C induces the decoupling of actin filament assembly and microtubule disassembly via relocating GEFH1 to the cytosol.

With this, the authors show that induction of TBC1D3C expression does in fact not affect the microtubule network while affecting GEFH1 localization. The same criticism as previously mentioned applies to fluorescence images.

Main Point 5. Direct interaction between TBC1D3C and GEF-H1

Looks good to me.

Main Point 6: Luan et al. used SIM imaging to assess the co-localization of GEFH1 with microtubules before and after Dox treatment.

The treatment condition without doxycycline induction displays clear co-localization of GEFH1, whereas there seems to be less co-localization in the dox-induced treatment. What, however, needs to be taken into account is that the images don't seem to be properly aligned in the dox condition, which becomes apparent when looking at the bottom edge of the image as well as the circle around the nucleus in the microtubule and GEFH1 condition. The authors should perform image registration, or at least manually align the separate images to properly assess the co-localization of the two channels.

While the live-cell imaging does provide some additional proof that GEFH1 dissociates from the microtubules and the microtubules stay intact all the while, however, when looking at the panels in figure 6D, for this reviewer, it is impossible to tell by eye whether some z-drift occurred as it appears that the nucleus is clearly visible in the first timepoints in the actin channel and while it stays visible in the other two channels the nucleus does not appear to be visible in the actin channel towards the end of the timelapse. I'd double-check that the correct planes are shown, and no z-drift occurred. The corresponding movies are also a bit unclear as they appear to show a z-stack and time combined in one; this is a bit confusing, and I would suggest separating the different planes below each other, which can be easily done using open-source software such as ImageJ. Additionally, the 3 channels should also be visible in the same movie as they clearly correspond to each other, and this makes direct comparison of the different channels very difficult.

General comments: Overall, what this reviewer finds mostly lacking are quantifications of various fluorescence images that would clearly strengthen the message of this paper. The western-blot data seems very clear and supportive of the main messages the authors want to make. The fluorescence data presented in Figure 6A (SIM imaging) needs to be aligned properly to correctly present colocalization data. The Live-cell imaging data needs to be double checked for corresponding image z-

planes and the movies should be formatted to present the data more clearly. As an additional comment, the colormaps chosen to represent the various fluorescence images are not ideal. As a rule, I would suggest switching out the colormaps to green, cyan, and magenta since these are easier to see and provide more contrast, especially when printed (where the blue color would essentially disappear). I would also suggest to the authors to include scale bars that are of higher resolution since the ones in figure 6a are barely legible, and the ones in figure 6b are all but absent. With the requested additional quantification of the IF data as well as the minor adjustments to the representation of the data I would recommend this for publication after a revision.

Reviewer #1 (Comments to the Authors (Required)):

Luan et al. "Decoupling actin assembly from microtubule disassembly by TBC1D3C-mediated direct GEF-H1 activation".

In this manuscript, the authors present the results of experiments designed to determine the effect of TBC1D3C on the actin cytoskeleton and the underlying mechanism. TBC1D3C is a Rab-family GAP lacking the catalytic arginine, therefore has little or no GAP activity. Its expression is restricted to primates. The first present IF data supporting a function in F-actin formation and stress fiber formation. They interpreted their results to indicate a mutant of TBC1D3C did not. They then present evidence that the effect is mediated by RhoA. A proteomic approach identified GEF-H1 as a potential target. Using cells with GEF-H1 knocked out, the effects on RhoA appeared mediated by this exchange factor. The authors interpret their results examining microtubules as TBC having no effect, but that TBC caused the dissociation of GEF-H1 from microtubules. In a series of experiments using both bacterially expressed protein and cells expressed in mammalian cells, they determine that there might be direct binding of RBD to the DH-PH tandem of GDF-H1. They then tried to corroborate their results using SIM and live cell imaging. Although they present an interesting idea, the results are not sufficiently developed to be published at this time. The major problems are 1. IF that is difficult to interpret 2. Lack of replicates 3. Lack of quantification 4. Insufficient sampling of cells to draw conclusions. The shortcomings are described in more detail below.

We really appreciate your critical evaluation and constructive feedback on our manuscript. We understand your reserved enthusiasm and fully agree to what you raised regarding the shortcomings of our manuscript. We have made substantial revisions to address your questions and improve the manuscript accordingly. We are grateful for the opportunity to address your valuable critiques and hope that you are satisfied with this revised manuscript.

Please find our detailed responses in the following paragraphs.

1. Differences in actin structures are not obvious in figure 1. Furthermore, actin stress fibers can be quantified from images and should be in this case. In addition, a sufficient number of cells need to be examined, as there is usually significant variability in actin structures within a single cell population.

Thank you for your insightful comments. We acknowledge the importance of accurately quantifying actin stress fibers to support our conclusions. Following your suggestions, we have now performed detailed quantifications of actin stress fibers, analyzing at least 50 cells per experiment. The results, showing the percentage of cells with actin stress fibers, have been presented in the revised Figure 1 to illustrate the effect of TBC1D3C on actin polymerization. Additionally,

the quantified data from three biological replicates (n=3) are included in the Source Data, which is provided as a Supplemental File. Figure 1a-c and Source data are shown as below.

Fig.1b							
	%cells with stress fibers						
	rep1	rep2	rep3	mean	SD	p value	
Dox 0h	15	19	10	14.66667	4.50925		
Dox 6h	25	30	34	29.66667	4.50925	0.015171639	
Dox 12h	55	40	42	45.66667	8.144528	0.004485206	
Dox 24h	74	67	80	73.66667	6.506407	0.000207681	
Dox 24h+CytD	7	8	10	8.333333	1.527525	7.13353E-05	
Dox 24h+Pha	74	76	80	76.66667	3.05505	0.509751088	

Fig.1c							
	%cells with stress fibers						
	rep1	rep2	rep3	mean	SD	p value	
Dox 0h	15	10	10	11.66667	2.886751		
Dox 6h	14	15	10	13	2.645751		
Dox 12h	16	13	10	13	3		
Dox 24h	15	17	8	13.33333	4.725816		
Dox 24h+CytD	16	16	10	14	3.464102		
Dox 24h+Pha	74	76	68	72.66667	4.163332	8.25598E-05	

- In figure 1B, only one experiment with no replicates and no quantification is shown, precluding interpretation.

Thank you for your sound critiques. We have now re-numbered the previous Figure 1B as Figure 1d and have revised it to include replicates and quantification to strengthen our conclusions. We have also quantified the levels of G-actin and

F-actin, as shown in Figures 1e and 1f, respectively. Additionally, we have also provided the relative ratio of F-actin to total actin from two biological replicates (n=2) in the Source Data, which is available as a Supplemental File. Figure 1d-f and Source data are shown as below.

Fig.1e						
Relative intensity						
	rep1	rep2		mean	SD	p value
Dox 0h	1.001884	0.998116		1	0.002664	
Dox 6h	3.458947	3.024222		3.241585	0.307397	0.009272933
Dox 12h	4.143392	3.863578		4.003485	0.197859	0.002163201
Dox 24h	3.939401	3.869926		3.904664	0.049127	0.000143416
Dox 24h+CytD	2.159921	1.732833		1.946377	0.301997	0.011986791
Dox 24h+Pha	3.26529	3.30911		3.2872	0.030985	0.085275098
Fig.1f						
Relative intensity						
	rep1	rep2		mean	SD	p value
Dox 0h	1.00648	0.99352		1	0.009164	
Dox 6h	0.807158	0.839093		0.823125	0.022582	
Dox 12h	0.859986	0.714768		0.787377	0.102684	
Dox 24h	1.29147	1.34077		1.31612	0.034861	
Dox 24h+CytD	0.858538	0.543077		0.700808	0.223065	0.041201506
Dox 24h+Pha	1.268656	1.27064		1.269648	0.001403	

- Quantification by mass spec is very difficult and the appropriate steps to accomplish this were not used.

Thank you for your insightful critique regarding our mass spectrometry

quantification. To clarify, we used Parallel Reaction Monitoring (PRM) analysis for the relative quantification of target peptides. We apologize for the lack of sufficient details in our previous description. In this revised manuscript, we have included detailed description of our use of PRM procedures in the Materials and Methods section. For your convenience, we also outlined them below:

For the PRM analysis, we selected unique peptides for our proteins of interest. The corresponding peptide information, including their retention time, m/z value, and ion charge, was obtained from Data-Dependent Acquisition (DDA) data. This precursor ion information was exported to an Excel file and used for PRM analysis. PRM samples were prepared similarly to DDA samples, with each group having four replicates. During PRM acquisition, the peptides of interest were selectively fragmented, and both their precursor and fragment ions were detected. A retention time window of ± 10 min was used to ensure peptide detection, and fewer than 30 peptides were analyzed simultaneously to maintain a reasonable cycle time. The acquired PRM data were directly imported into Skyline to quantify the precursor ions with their three isotopic peaks and the top five fragment ions (b ions and y ions). The related peptide information was then exported, and the peak areas of the fragment ions were summed up as the peptide signal intensity. Finally, the selected peptides were integrated as protein levels.

4. In figure 2A, the total amount of RhoA is changing. There is no normalization so can't be interpreted. In addition, a single experiment, no replicates, no quantification precluding interpretation.

Thank you for your valuable suggestions. We agree with your concerns that proper quantification and normalization are essential in this case when drawing accurate conclusions. To address these concerns, we have performed quantification of active and total RhoA in Figure 2a and presented the normalized data in Figures 2b and 2c, now based on two biological replications (n=2). The associated relative quantification values are included in the Source Data, which is provided as a Supplemental File. Figure 2a-c and Source data are shown as below.

Fig.2b						
Relative intensity						
	rep1	rep2	mean	SD	p value	
Dox 0h	1.000733	1.008338	1.004536	0.005378		
Dox 6h	1.853792	2.107845	1.980818	0.179643	0.008263	
Dox 12h	3.997452	2.112363	3.054908	1.332959	0.0808	
Dox 24h	3.893001	2.384343	3.138672	1.066782	0.052764	
Dox 24h+Noc	3.208455	1.357956	2.283205	1.308501	0.274	
Fig.2c						
Relative intensity						
	rep1	rep2	mean	SD	p value	
Dox 0h	1.001566	1.002464	1.002015	0.000635		
Dox 6h	1.187039	1.405219	1.296129	0.154276	0.114438	
Dox 12h	2.211824	1.4104	1.811112	0.566692	0.180922	
Dox 24h	2.008645	1.419249	1.713947	0.416766	0.136999	
Dox 24h+Noc	5.557579	3.124546	4.341062	1.720414	0.085347	

5. On line 152, I think actually referring to fig 3, not fig 2 as written. Some problems as with comment 1.

Thank you for your careful review and attention to detail. We have revised our text to clarify any potential confusion.

In Figure 2, we were to demonstrate that RhoA mediates actin filament assembly induced by TBC1D3C. As detailed in our response to Comment 4, we performed immunoblotting analysis of active RhoA induced by TBC1D3C overexpression, which is shown in Figure 2a. We quantified their levels in Figures 2b and 2c. We validated this observation by using C3 exoenzyme and Rhosin, as shown in Figures 2d-g, further supporting that TBC1D3C stimulates F-actin polymerization via RhoA.

In Figure 3, we aimed to demonstrate that the activation of GEF-H1 mediates TBC1D3C-induced RhoA activation. This demonstrates another level of the mechanisms by which TBC1D3C regulates actin dynamics, complementing the data shown in Figure 2.

For both Figures 2 and 3, we have performed proper quantification on all relevant figure panels to better support our conclusions in this revised manuscript.

We appreciate the opportunity to clarify these points and hope that our revisions meet your expectations.

6. Fig 3A is a single experiment, no replicates, no quantification, no normalization, precluding interpretation, and mutant data not clear.

Thank you for pointing out the shortcomings. We agree with you that a single experiment without replicates and quantification is insufficient to draw reliable conclusions. To address this, we have quantified active and total GEF-H1 in Figure 3a and presented the normalized data (total GEF-H1 expression levels were used to normalize the levels of active GEF-H1) in Figures 3b-c.

Regarding the TBC1D3C mutant data, we have also performed quantification and normalization of active GEF-H1 to confirm its inability to induce GEF-H1 activation.

All the associated quantification data (n=2, from two biological replicates) are now included in the Source Data, provided as a Supplemental File. These additions ensure proper quantification and normalization, enhancing the interpretability of our results. Figure 3a-c and Source data are shown as below.

Fig.3b						
Relative intensity						
	rep1	rep2	mean	SD	p value	
Dox 0h	1.063119	0.936881		1	0.089263	
Dox 6h	2.303073	2.111907	2.20749	0.135174	0.004439	
Dox 12h	3.362219	2.126393	2.744306	0.873861	0.053429	
Dox 24h	6.939223	5.992285	6.465754	0.669586	0.003775	
Dox 0h+Noc	8.630344	11.9335	10.28192	2.335683	0.015137	
Fig.3c						
Relative intensity						
	rep1	rep2	mean	SD	p value	
Dox 0h	1.592658	0.407342		1	0.838146	
Dox 6h	0.587509	1.897294	1.242402	0.926158	0.404747	
Dox 12h	1.131791	1.423246	1.277519	0.20609	0.34695	
Dox 24h	2.970521	0.663231	1.816876	1.6315	0.296582	
Dox 24h+Noc	10.2333	11.96179	11.09754	1.222228	0.011642	

7. In line 186, state no alteration in microtubules, but the conclusion is not accurate because there might be a change in modification of tubulin. It is very difficult to detect changes in microtubule morphology in cells. Dynamics also were not examined.

Thank you for your sound critique. We apologize for the inaccurate description in our manuscript. To clarify, our intention was to show that microtubules are not depolymerized upon TBC1D3C expression. We have modified our description in this revised version to avoid any potential confusion.

Regarding microtubule dynamics, we have included live-cell imaging data in Figure 6, demonstrating the dynamics in microtubules, actin, and GEF-H1 with increased induction of TBC1D3C.

8. GEF-H1 distribution results have the some shortcoming described above for the other data

Thank you for your insightful comment regarding our GEF-H1 distribution results. We acknowledge the shortcomings in our previous version and have now rectified them in this revised manuscript. To address this, we have now included an analysis of the colocalization between GEF-H1 and microtubules by calculating their correlation coefficients, as shown in Figures 4d-e. This quantitative approach provides a more robust assessment of the alterations in GEF-H1 distribution. The associated quantification data (n=7) have been included in the Source Data, which is provided as a Supplemental File. Figure 4c-e and Source data are shown as below.

Fig.4d											
Correlation coefficient											
	rep1	rep2	rep3	rep4	rep5	rep6	rep7		mean	SD	p value
Dox 0h	0.6411	0.7	0.58	0.72	0.65	0.68	0.6		0.67055	0.041649	
Dox 24h	0.4836	0.45	0.4	0.5	0.42	0.48	0.44		0.4668	0.023759	4.47981E-06
Dox 0h+Noc	0.42	0.43	0.4	0.45	0.5	0.42	0.52		0.425	0.007071	4.48472E-06
Dox 24h+Noc	0.549	0.5	0.52	0.44	0.56	0.48	0.45		0.5245	0.034648	0.057298373
Dox 24h+PTX	0.3182	0.35	0.4	0.42	0.45	0.34	0.36		0.3341	0.022486	0.005962979

Fig.4e											
Correlation coefficient											
	rep1	rep2	rep3	rep4	rep5	rep6	rep7		mean	SD	p value
Dox 0h	0.6079	0.62	0.7	0.64	0.6	0.71	0.66		0.61395	0.008556	
Dox 24h	0.5489	0.57	0.62	0.7	0.61	0.65	0.73		0.55945	0.01492	0.611189688
Dox 0h+Noc	0.497	0.48	0.5	0.44	0.46	0.52	0.43		0.4885	0.012021	2.46027E-06
Dox 24h+Noc	0.55	0.5	0.44	0.52	0.46	0.53	0.47		0.525	0.035355	0.000515429

9. In both SIM and live cell experiments, conclusions are based on single cells. See comment 1.

Thank you for your sound critique. We agree that single-cell results are insufficient for drawing reliable conclusions. To address this, we have performed quantification analysis on multiple cells for both SIM and live-cell experiments. Specifically, for the SIM data in Figure 6a, we quantified results from five cells and calculated the correlation coefficient of colocalization between GEF-H1 and microtubules (Figure 6b), as well as quantified F-actin stress fibers (Figure 6c). For the live-cell experiments (Figure 6d and associated live-cell imaging video), we also quantified F-actin stress fibers across eight time-intervals from five cells (Figure 6e). The associated quantification data (n=5) are now included in the Source Data, provided as a Supplemental File. We believe these revisions ensure a more robust and statistically reliable interpretation of our findings. Figure 6a-e and Source data are shown as below.

Fig.6b								
Correlation coefficient								
	rep1	rep2	rep3	rep4	rep5	mean	SD	p value
Dox 0h	0.7135	0.68	0.65	0.72	0.7	0.69675	0.023688	
Dox 24h	0.2959	0.33	0.35	0.4	0.2	0.31295	0.024112	0.000112
Fig.6c								
Relative intensity								
	rep1	rep2	rep3	rep4	rep5	mean	SD	p value
Dox 0h	1.259491	1.149852	0.746195	0.595526	1.248936	1	0.30813	
Dox 24h	1.80224	2.446798	1.762232	2.542988	2.253396	2.161531	0.361889	0.000652
Fig.6e								
Relative intensity								
	rep1	rep2	rep3	rep4	mean	SD	p value	
Dox 0 min	0.891097	1.16461	1.229082	0.71521	1	0.239819		
Dox 60 min	1.135564	1.476703	1.366022	0.998185	1.244118	0.216967	0.182345	
Dox 120 min	1.140332	1.310791	1.311305	0.964151	1.181645	0.165832	0.264694	
Dox 240 min	1.361328	1.575209	1.200183	0.854864	1.247896	0.303727	0.123715	
Dox 360 min	2.318071	2.267168	1.369836	1.111506	1.766645	0.616781	0.029848	
Dox 480 min	2.043092	2.817053	1.99835	1.711048	2.142386	0.473228	0.002528	
Dox 600 min	2.66295	1.683543	1.659411	1.204878	1.802696	0.614316	0.02543	
Dox 720 min	1.565087	2.587183	2.495938	1.93131	2.144879	0.483269	0.002707	

Minor: the authors don't explain why they selected the particular mutants of TBC to examine.

Thank you for your insightful comment. We selected the specific mutants of TBC1D3C, Arg107 and Asp148, because these are conserved residues within the TBC domain, which plays a crucial role in TBC1D3C's function. Previous studies have demonstrated that mutations in these conserved amino acids can abolish the carcinogenic ability of TBC1D3C, suggesting their importance in the protein's activity. Therefore, we hypothesized that these mutants might also affect the role of TBC1D3C in actin polymerization. Our pilot tests confirmed that these mutants lost their activity in inducing GEF-H1 activation and promoting F-actin polymerization. Based on this, we used them side-by-side with the wild-type TBC1D3C as ideal controls to better understand the function of TBC1D3C in our experiments.

In short, although interesting, the work is at a preliminary stage and publication is not warranted.

Thank you for your thoughtful feedback and potential enthusiasm in our work. We have carefully addressed all the aspects you pointed out in your review, incorporating your valuable suggestions to enhance the rigor and clarity of our study. We believe these revisions have significantly strengthened our study, as outlined below:

1) Data Quantification and Statistical Analysis: We have conducted extensive quantification of actin stress fibers, Western blots, and colocalization coefficients, which are now included in figures such as Figures 1b-c, e-f, Figures 2b-c, Figures 3b-f, Figures 4d-e, and Figures 6b-e. Additionally, we have performed statistical

analyses on all relevant datasets and updated both the figures and textual content throughout the manuscript to reflect these enhancements.

- 2) **Expanded Experimental Descriptions:** We have included detailed descriptions of key experimental procedures, such as actin segmentation, microtubule co-sedimentation assays, and peptide quantification by mass spectrometry in the Materials and Methods section. These additions provide greater transparency and reproducibility.
- 3) **Enhanced Imaging and Data Presentation:** We have refined the alignment of the fluorescence data in Figure 6a (SIM imaging) and the live-cell imaging data in Figure 6d and associated videos for improved presentation. High-resolution scale bars have also been added to the images in Figure 6a.
- 4) **Clarification of Key Concepts:** We have provided a more detailed explanation of the concept of “decoupling,” the rationale behind studying TBC1D3C, and the potential mechanisms of TBC1D3C activation. These clarifications help contextualize our findings within the broader scientific landscape.
- 5) **Methodological Clarifications:** We have clarified the methodologies used for affinity purification and subsequent mass spectrometry identification of RhoA and GEF-H1. Additionally, we have provided more details on the construction of the GEF-H1 knockout cell line using the CRISPR/Cas9 system, ensuring a comprehensive understanding of our experimental approach.

These revisions have addressed your concerns and demonstrated that our work has progressed beyond a preliminary stage. Our study offers novel insights into the molecular mechanisms underlying dynamic interactions between actin and microtubules exemplified by TBC1D3C's regulation, which we believe will contribute valuable knowledge to the field. We hope you will find our revisions satisfactory and supportive of its publication.

Reviewer #2 (Comments to the Authors (Required)):

This manuscript by Luan et al. describes the role of TBC1D3C in GEF-H1-mediated RhoA activation. The authors argue that actin and microtubule dynamics are decoupled and that TBC1D3C can have an important role in this process. The manuscript certainly has merits but the data provided does not fully support the title of the manuscript "decoupling actin from microtubule disassembly by TBC1D3C-mediated direct GEF-H1 activation". It is also not clear what the authors mean with "decoupling", does it mean that actin and microtubule is occurring simultaneously and that TBC1D3C somehow disrupts this coordination? The authors need to be more specific about what they mean. It is not clear why the authors started to study TBC1D3C. If it is only expressed in hominoids, it cannot be such a global factor in the communication between actin and microtubule assembly. What about this process in other species? Have all proteins with RabGAP domains a similar function? Do U2OS cells express TBC1D3C? The authors have created stable a cell line with inducible expression of TBC1D3C, i.e. what they study is an alteration in changes in steady state conditions over time, not a dynamic process. How is TBC1D3C activated under normal conditions? It is not clear how the observation from this lab-generated model system compares to the in vivo situation. With this said, I think the study is in most aspects well done but there are several weak points that will need some input from the authors.

Thank you for your constructive and comprehensive comments on our manuscript. Your insightful comments have been instrumental in improving our study. We appreciate the opportunity to clarify and strengthen our study. Below, please find our responses to your key concerns and provide additional context for our findings.

1) Decoupling Actin assembly from Microtubule Disassembly: By "decoupling," we refer to the ability of TBC1D3C to activate GEF-H1-RhoA cascade and promote actin filament formation independently of microtubule disassembly. While the dynamic assembly and disassembly of actin and microtubules are often coupled, TBC1D3C can specifically regulate actin dynamics without affecting microtubule stability. This suggests a unique regulatory role for TBC1D3C in modulating cytoskeleton dynamics.

2) Rationale for Studying TBC1D3C: At one point, we were to study putative effects of the deubiquitination enzymes (DUBs) and their related proteins on cytoskeleton dynamics via overexpression studies. By serendipity, we noticed an interesting effect of TBC1D3C on actin and microtubule dynamics. Although TBC1D3C is exclusive to hominoids, we thought its unique feature of decoupling actin and microtubule dynamics can provide critical insights into novel cellular mechanisms in regulating cytoskeleton dynamics; so, we decided to split some of our efforts to study and publish it to the community, which may provide invaluable insights into potentially novel, hominoid-specific cellular mechanisms and

evolutionary adaptations in cytoskeleton regulation. While we have not performed studies on the broader implications for other species and similar functions of other RabGAP domain-containing proteins, which are interesting and promising areas for future studies, our findings from TBC1D3C offer valuable insights and lay a good foundation for future related studies.

3) Expression and Activation of TBC1D3C Using Inducible System in U2OS cells:

U2OS cells do not express TBC1D3C based on our detection using RT-qPCR and immunoblotting analyses. Additionally, we could not find any cell lines or tissues with detectable TBC1D3C (or other TBC1D3 members), despite extensive searches in public gene and protein expression databases and our own tests over many other cell lines. Considering TBC1D3C can be a good example to reveal novel insights about cytoskeleton dynamics, we opted to generate an inducible U2OS cell line to express TBC1D3C to facilitate controlled studies of its regulatory mechanism on decoupling actin polymerization from microtubule depolymerization.

Regarding the upstream regulators of TBC1D3C under normal conditions, it remains largely undefined. However, a yeast two-hybrid screening study identified potential binding partners for TBC1D3C, including nischarin. Nischarin is known to interact with the cytoplasmic domain of integrin $\alpha 5$, inhibiting cell motility and altering actin organization. We hypothesize that external stimuli might lead to the release of nischarin via $\alpha 5 \beta 1$ integrin, thereby activating TBC1D3C. This hypothesis presents worth for further studies when cells with endogenous TBC1D3C expression are available.

4) Model System and *In Vivo* Comparisons: Given the absence of cell lines or tissues with detectable endogenous TBC1D3C expression, we used this inducible U2OS cell model system as a novel approach to investigate its regulation on cytoskeleton dynamics.

We acknowledge that our findings in this artificial system may not capture *in vivo* complexities entirely, although this system may provide valuable insights into the potential role of TBC1D3C and into novel mechanism of cytoskeleton dynamics. Future studies will aim to explore the physiological relevance and regulation of TBC1D3C under normal conditions when cells and tissues with endogenous TBC1D3C expression are available.

We hope these clarifications address the key concerns and enhance the overall context of our findings. Thank you again for your valuable feedback.

Detailed comments:

1. line 123 and Fig.1B. The authors describe the increase in F-actin and a corresponding decrease. It is not described in the Mat&Meth section. What is the buffer? Conditions for ultracentrifugation? The authors do not relate the amount of

G-actin to the amount of total actin (i.e. before centrifugation). I also suggest that the authors make an effort to quantify the stress fiber formation in Fig.1A.

Thank you for your valuable suggestions. We apologize for the omission of details regarding the G-actin and F-actin separation method. We have now revised the Materials & Methods section to include details of the procedures and buffers used.

Revised procedure for actin sedimentation by centrifugation is as follows:

Actin segmentation by centrifugation

Cells were homogenized in RIPA buffer (#P0013B, Beyotime, China) supplemented with 1% phenylmethanesulfonylfluoride (#ST506, Beyotime) for 30 min on ice before being collected. The lysates were then subjected to centrifugation at $15,000 \times g$ for 30 min at 4 °C. The supernatant containing G-actin was transferred to a fresh tube, while the pellet containing F-actin was resuspended in cold PBS and then centrifuged at $15,000 \times g$ for 5 min twice. After centrifugation, the pellet was resuspended in F-actin extracting solution (1.5 mM guanidine hydrochloride, 1 mM sodium acetate, 1 mM CaCl_2 , 1 mM ATP, 20 mM Tris-HCl, pH 7.5) and incubated for 1 h on ice to dissolve F-actin before being centrifuged at $15,000 \times g$ for 30 min at 4 °C. Both forms of actin were finally analyzed via immunoblotting against β -actin.

Additionally, we have now quantified actin stress fiber formation in Figure 1a, showing in Figure 1b-c, to demonstrate the effect of TBC1D3C on actin polymerization. We have also quantified the levels of G-actin and F-actin shown in Figure 1d and presented them in Figure 1e-f. All the associated quantification data are included in the Source Data, supplied as a Supplemental File. Figure 1 and Source data are shown as below.

2. line 132. The authors claim that they have made an unbiased affinity purification. I do not agree, since there is a bias for the Rho subgroup. What about Rac and Cdc42? Alterations in their activities could also affect the status of actin.

Thank you for your insightful feedback regarding our affinity purification approach and our wording in the description. We apologize for the oversight in describing our affinity purification as unbiased and have modified it in this revised manuscript. Before conducting our GST-rhotekin-Rho binding domain (GST-RBD) affinity purification and mass spectrometry identification, we performed pilot experiments using C3 exoenzyme to narrow down the involvement of specific small GTPase subfamilies. C3 is well-known for its specific inhibitory activity against members of the Rho family of small GTPases, such as RhoA, RhoB, and RhoC, but it does not typically affect other members like Rac or Cdc42. Our results indicated that C3 treatment abolished the effects of TBC1D3C-induced F-actin formation. Given the specificity of C3 targeting Rho proteins, we therefore used GST-RBD to specifically capture activated Rho GTPases and use mass spectrometer to identify which members were activated by TBC1D3C, which turned out to be mainly RhoA. We later used data from both C3 and Rhosin to corroborate our findings, confirming that TBC1D3C

predominantly regulates RhoA without much effects on Rac or Cdc42 (Figures 2d-g).

3. Line 162, the authors found an accumulation of GEF-H1 but it is not clear if there were other RhoGEFs in the precipitate and if GEF-H1 was cherry picked because it followed the hypothesis.

Thank you for your thoughtful comments. In addition to GEF-H1, our mass spectrometry analysis also identified a slight accumulation of ARFGEF1 in the precipitate. It is important to note that ARFGEF1 predominantly functions as a GEF for ARF family members, with minimal activity towards RhoA. In contrast, GEF-H1 is directly involved in the regulation of RhoA activity. We therefore focused on analyzing GEF-H1 accumulation in response to TBC1D3C overexpression.

4. Line 171, It is not clear what the authors mean by GEF-H1 activation? Activation in what sense? Increased binding?

Thank you for your insightful question. By "GEF-H1 activation," we refer to the process by which GEF-H1 transitions from an inactive state, where it is normally bound to microtubules in an inhibited and inactive conformation, to an active state, where it is released into the cytosol, undergoes a conformational change and becomes capable of interacting with and activating RhoA, thereby promoting actin stress fiber formation.

5. Extended data figures does that mean supplementary figures? This is a bit confusing.

Thank you for highlighting this issue. To clarify, "Extended Data Figures" refers to what are commonly known as supplementary figures. We have revised the manuscript to use the term "supplementary figures" for clarity.

6. Fig.3B and the panels showing cells onward. Here is the problem with showing single cells. How do we know that they are representative?

Thank you for raising this important point. We agree that single-cell images, although illustrative, may not be sufficient for robust conclusions. To address this concern, we have expanded our analysis in Figure 3d (i.e., the original Figure 3b) to include quantification of actin stress fiber formation across a larger number of cells, which is shown in Figures 3e-f. The percentage of cells exhibiting actin stress fibers, based on three biological replicates (n=3), is provided in the Source Data available as a Supplemental file.

Similarly, for other panels showing cells across all the figures, we have included multiple cells and performed quantifications to ensure them to be representative. Figure 3d-f and Source data are shown as below.

Fig.3e							
%cells with stress fibers							
	rep1	rep2	rep3		mean	SD	p value
Dox 0h	12	23	16		17	5.567764	
Dox 6h	37	33	34		34.66667	2.081666	0.006755
Dox 12h	55	52	43		50	6.244998	0.002401
Dox 24h	71	68	75		71.33333	3.511885	0.000139
Dox 0h+Noc	82	78	68		76	7.211103	0.00036

Fig.3f							
%cells with stress fibers							
	rep1	rep2	rep3		mean	SD	p value
Dox 0h	14	12	19		15	3.605551	
Dox 6h	16	15	11		14	2.645751	0.71826
Dox 12h	15	13	12		13.33333	1.527525	0.501899
Dox 24h	15	14	7		12	4.358899	0.410304
Dox 0h+Noc	77	68	81		75.33333	6.658328	0.000161

7. Fig 4B. The microtubule staining is very difficult to follow in these minuscule sub panels. Would it be possible to show magnified section of some of the key conditions?

Thank you for your constructive feedback. We have addressed your concern by magnifying the microtubule staining panels in Figure 4b to provide a clearer and more detailed view of the microtubules. We hope this enhancement improves the interpretability of the results. Figure 4b is shown as below.

8. Fig 4C. Again, the low resolution makes it difficult to evaluate the alleged effects on GEF-H1 relocalization. I suggest that the authors perform co-localization experiments in a more quantitative manner.

Thank you for your valuable feedback. Following your suggestion, For Figure 4c, we have performed quantitative co-localization analysis to better assess GEF-H1 relocalization. Specifically, we measured the co-localization coefficient between GEF-H1 and microtubules to support our hypothesis that TBC1D3C induces GEF-H1 release from microtubules, resulting in its more uniform distribution throughout the cytoplasm without altering microtubule morphology, as shown in Figures 4d-e. The associated detailed results of these measurements (n=7) are now included as source data in the Supplemental file. Figure 4c-e and Source data are shown as below.

Fig.4d										
Correlation coefficient										
	rep1	rep2	rep3	rep4	rep5	rep6	rep7	mean	SD	p value
Dox 0h	0.6411	0.7	0.58	0.72	0.65	0.68	0.6	0.67055	0.041649	
Dox 24h	0.4836	0.45	0.4	0.5	0.42	0.48	0.44	0.4668	0.023759	4.47981E-06
Dox 0h+Noc	0.42	0.43	0.4	0.45	0.5	0.42	0.52	0.425	0.007071	4.48472E-06
Dox 24h+Noc	0.549	0.5	0.52	0.44	0.56	0.48	0.45	0.5245	0.034648	0.057298373
Dox 24h+PTX	0.3182	0.35	0.4	0.42	0.45	0.34	0.36	0.3341	0.022486	0.005962979

Fig.4e										
Correlation coefficient										
	rep1	rep2	rep3	rep4	rep5	rep6	rep7	mean	SD	p value
Dox 0h	0.6079	0.62	0.7	0.64	0.6	0.71	0.66	0.61395	0.008556	
Dox 24h	0.5489	0.57	0.62	0.7	0.61	0.65	0.73	0.55945	0.01492	0.611189688
Dox 0h+Noc	0.497	0.48	0.5	0.44	0.46	0.52	0.43	0.4885	0.012021	2.46027E-06
Dox 24h+Noc	0.55	0.5	0.44	0.52	0.46	0.53	0.47	0.525	0.035355	0.000515429

9. 6A the superresolution experiments are not really convincing. I suggest that the authors use another color than blue for microtubules (in 6A and B).

Thank you for your constructive suggestion. We agree that the blue color used in the previous version may not be in optimal contrast for visualizing morphological alterations in microtubules. We have now revised Figures 6a and 6d (formerly Figures 6A and 6B) by changing the microtubule labeling color to magenta. This change enhances the clarity and visibility of the microtubule structures. Figure 6a-e is shown as below.

10. Fig. 6C It is not clear how the sedimentation assay has been done. Gentrification, how?

Thank you for bringing out this point. We apologize for the lack of details in the previous version. We have now included detailed procedure for the sedimentation assay in this revised manuscript, which is also shown below for your convenient reference:

Microtubule Co-sedimentation Assay:

Cells were washed twice with PBS and then extracted using microtubule-stabilizing buffer (100 mM PIPES, pH 6.9, 5 mM MgCl₂, 2 mM EGTA, 2 M glycerol, 0.1% NP40, 10 mM β-glycerophosphate, 50 mM NaF, 0.3 mM okadaic acid (Roche, IN), 1 mM PMSF and protease inhibitor cocktail) for 15 minutes at room temperature. The extract was then centrifuged at 1,000 × g for 5 min at room temperature, which is sufficient to pellet both the unextractable and the microtubule-enriched cytoskeleton components. The resulting supernatant contains the soluble cytosolic fraction, including the majority of unassembled tubulin.

11. Fig.5E I suggest that the authors also include a figure of the domain organization of TBC1D3C. How many amino acid residues in GEF-H1 and TBC1D3C?

Thank you for your valuable suggestion. We have now included the domain organization of TBC1D3C in Figure 5e and provided the number of amino acid residues for both GEF-H1 and TBC1D3C. Figure 5e is shown as below.

Reviewer #3 (Comments to the Authors (Required)):

Overview: in their work, Luan et al. present "Decoupling actin assembly from microtubule disassembly by TBC1D3C-mediated direct GEF-H1 activation" a mechanism by which GEF-H1 mediated RhoA activation can be accomplished independently of the state of the microtubule array. They propose a method by which TBC1D3C can bind GEF-H1, thereby disrupting its coupling to the Tctex-DIC-14-3-3 complex and displacing it into the cytosol where it can get activated and promote RhoA activation, which ultimately leads to the assembly of filamentous actin and stress-fiber formation.

Their work consists of measuring RhoA activity in cells expressing a doxycycline-inducible variant of TBC1D3C, where they show that induction leads to a significant increase in active RhoA in U2OS cells. The Authors show that this corresponds to an increase in filamentous actin as measured using western blots and phalloidin stainings at different timepoints after induction of TBC1D3C expression (Figures 1 and 2). Next, Luan et al. present results that illustrate the increase of active GEFH1 in their Tet-inducible system correlated with an increase in cytosolic GEFH1 as imaged using transfected GEFH1 in immunofluorescence experiments. A knockout of GEF-H1 seems to fully abolish active RhoA and leads to less f-actin as shown using phalloidin staining (Figure 3). This seems to be independent of the state of the microtubule cytoskeleton, as GEFH1 could get released from microtubules without the microtubules getting disrupted. Additionally, the expression of TBC1D3C disrupts the interaction of GEF-H1 with the Tctex-1DIC-DHC complex, which suggests that TBC1D3C directly binds GEFH1 and leads to its release from microtubules. A series of truncation and deletion experiments suggest that GEFH1 DH-PH domain interacts with the N-terminal domain of TBC1D3C.

Thank you very much for your insightful and constructive comments on our manuscript. We greatly appreciate your thorough review and the opportunity to address your valuable critiques. In response to your feedback, we have made substantial revisions to the manuscript to address your concerns and improve its clarity and accuracy. We hope these changes meet your expectations. Please find our detailed responses in the following paragraphs.

Main point 1: TBC1D3C promotes F-actin filament formation. Here the authors used their TBC1D3C inducible cell line to test for an increase in F-actin formation during induction of protein expression. They provide clear results using ultracentrifugation, showcasing an increase in F-actin formation after expression induction. They also show IF images labeled with phalloidin. The way the fluorescence data is presented now might provide excerpts of individual cells; however, quantification of this is required to clearly validate this result. A very straightforward way to assess this would be to simply quantify mean intensity per cell in a series of fields of view. This simple and straightforward quantification will strengthen the result.

Thank you for your valuable suggestions. We agree with you the importance of

quantifying actin stress fibers to substantiate our conclusions. Following your recommendation, for Figure 1, we have now conducted detailed quantifications of actin stress fibers, analyzing at least 50 cells per experiment. We have also quantified the percentage of cells exhibiting actin stress fibers, demonstrating the effect of TBC1D3C on actin polymerization.

Furthermore, we have performed similar quantitative approaches to other figures when relevant, ensuring consistency across our analyses.

All quantified data (n=3 from three biological replicates, or n=2 from two biological replicates) are included in the Source Data, which is provided as a Supplemental File. Figure 1a-c and Source data are shown as below.

Fig.1b							
	%cells with stress fibers						
	rep1	rep2	rep3	mean	SD	p value	
Dox 0h	15	19	10	14.66667	4.50925		
Dox 6h	25	30	34	29.66667	4.50925	0.015171639	
Dox 12h	55	40	42	45.66667	8.144528	0.004485206	
Dox 24h	74	67	80	73.66667	6.506407	0.000207681	
Dox 24h+CytD	7	8	10	8.333333	1.527525	7.13353E-05	
Dox 24h+Pha	74	76	80	76.66667	3.05505	0.509751088	
Fig.1c							
	%cells with stress fibers						
	rep1	rep2	rep3	mean	SD	p value	
Dox 0h	15	10	10	11.66667	2.886751		
Dox 6h	14	15	10	13	2.645751		
Dox 12h	16	13	10	13	3		
Dox 24h	15	17	8	13.33333	4.725816		
Dox 24h+CytD	16	16	10	14	3.464102		
Dox 24h+Pha	74	76	68	72.66667	4.163332	8.25598E-05	

Main point 2: RhoA as the mediator of TBC1D3C-induced filament assembly. Here, the authors pinpoint RhoA as the mediating factor of increased filament assembly, which they again showcase using western blotting and immunofluorescence. The RBD pulldown assay of RhoA provides evidence that active RhoA does increase with increased expression time of TBC1D3C. Similarly, to the point above, quantification of fluorescence intensity would significantly strengthen the observed results. Additionally, what would also provide increased evidence is to measure active RhoA with fluorescence biosensors. Such a strong increase in RhoA activity as measured using the RBD pulldown would be clearly visible using any of the common RhoA activity reporters. However, since these could potentially be challenging experiments, I would not say that I request this experiment strictly, although in my opinion it would significantly strengthen the results.

Thank you for your constructive suggestions. To strengthen our evidence that active RhoA increases with TBC1D3C expression, we have quantified the relative intensity of active RhoA in Figure 2a. Additionally, we have included the quantification of active RhoA in Figures 2b and 2c. Our data indicate a clear increase in active RhoA following TBC1D3C induction, supporting our hypothesis. The relative intensity of active RhoA (n=2, 2 biological replicates) is provided in the Source data, available as a Supplemental file.

We really appreciate your suggestion to use fluorescence biosensors for measuring active RhoA. While we acknowledge the potential of such biosensors to provide dynamic and visual insights into RhoA activity, we were unable to include these experiments in the current study due to resource constraints. We sure will consider this approach a valuable direction in our future follow-up studies.

We hope you are satisfied that these revisions adequately demonstrate the relationship

between TBC1D3C expression and RhoA activation. Figure 2a-c and Source data are shown as below.

Fig.2b						
Relative intensity						
	rep1	rep2	mean	SD	p value	
Dox 0h	1.000733	1.008338	1.004536	0.005378		
Dox 6h	1.853792	2.107845	1.980818	0.179643	0.008263	
Dox 12h	3.997452	2.112363	3.054908	1.332959	0.0808	
Dox 24h	3.893001	2.384343	3.138672	1.066782	0.052764	
Dox 24h+Noc	3.208455	1.357956	2.283205	1.308501	0.274	
Fig.2c						
Relative intensity						
	rep1	rep2	mean	SD	p value	
Dox 0h	1.001566	1.002464	1.002015	0.000635		
Dox 6h	1.187039	1.405219	1.296129	0.154276	0.114438	
Dox 12h	2.211824	1.4104	1.811112	0.566692	0.180922	
Dox 24h	2.008645	1.419249	1.713947	0.416766	0.136999	
Dox 24h+Noc	5.557579	3.124546	4.341062	1.720414	0.085347	

Main Point 3: GEFH1 as the mediator between RhoA activation and TBC1D3C. Here the authors provide evidence that GEF-H1 is the connection between active RhoA and TBC1D3C. Using the GST-RhoA-G17A mutant, which binds preferentially to GEFs, after 24h of doxycycline-induced expression of TBC1D3C, they see a strong signal in the pulldown western blot of GEFH1. Additionally, using transiently expressed GEFH1 and phalloidin, they show that GEFH1 dissociates from microtubules upon induction of TBC1D3C expression, similar to what treatment with nocodazole yields. Again, quantification of the fluorescence data would provide more evidence complementing the qualitative nature of the fluorescence data currently provided. Secondly, for the knockout experiments using CRISPR-Cas, it seems that the authors used a transiently expressing cell line.

In my opinion it does not really make sense to use a transiently expressing cell line to showcase the validity of the knockout, at least not without quantifying the number of cells that show fluorescence, as the expression is transient anyway, and many cells also do not express the GEFH1 construct, which essentially renders the first panels in the knockout experiment not very informative. I would suggest that either the authors remove the top panels showing the transiently expressed GEFH1 data, provide some quantitative information about the number of cells expressing the construct, or probably best would be to repeat the experiments for a cell line expressing GEFH1 stably; otherwise, the information content is very limited since, like this, it does not provide additional information as the expression is anyway transient and thus an absence of fluorescence in single cells does not prove that the knockout was successful.

Thank you for your valuable suggestions. We agree with you that quantification of the fluorescence data would provide stronger support for the qualitative data presented. For Figure 3d, we have quantified the percentage of cells with actin stress fibers to demonstrate the effect of TBC1D3C on actin polymerization, as shown in Figure 3e-f. The associated quantification data, based on three biological replicates (n=3), are included in the Source Data, provided as a Supplemental File.

Regarding the GEF-H1 knockout experiments, we apologize for not describing it clearly. To clarify, the cells were stable GEF-H1 knockout, which were generated using the CRISPR/Cas9 system and selected with puromycin in 293A cells. The stable knockout efficiency of GEF-H1 was detected by immunoblotting as shown in Figure 3g. Figure 3 and Source data are shown as below.

Fig.3b						Fig.3e						
Relative intensity						%cells with stress fibers						
	rep1	rep2	mean	SD	p value		rep1	rep2	rep3	mean	SD	p value
Dox 0h	1.063119	0.936881	1	0.089263		Dox 0h	12	23	16	17	5.567764	
Dox 6h	2.303073	2.111907	2.20749	0.135174	0.004439	Dox 6h	37	33	34	34.66667	2.081666	0.006755
Dox 12h	3.362219	2.126393	2.744306	0.873861	0.053429	Dox 12h	55	52	43	50	6.244998	0.002401
Dox 24h	6.939223	5.992285	6.465754	0.669586	0.003775	Dox 24h	71	68	75	71.33333	3.511885	0.000139
Dox 0h+Noc	8.630344	11.9335	10.28192	2.335693	0.015137	Dox 0h+Noc	82	78	68	76	7.211103	0.00036

Fig.3c						Fig.3f						
Relative intensity						%cells with stress fibers						
	rep1	rep2	mean	SD	p value		rep1	rep2	rep3	mean	SD	p value
Dox 0h	1.592658	0.407342	1	0.838146		Dox 0h	14	12	19	15	3.605551	
Dox 6h	0.587509	1.897294	1.242402	0.926158	0.404747	Dox 6h	16	15	11	14	2.645751	0.71826
Dox 12h	1.131791	1.423246	1.277519	0.20609	0.34695	Dox 12h	15	13	12	13.33333	1.527525	0.501899
Dox 24h	2.970521	0.663231	1.816876	1.6315	0.296582	Dox 24h	15	14	7	12	4.358899	0.410304
Dox 24h+Noc	10.2333	11.96179	11.09754	1.222228	0.011642	Dox 0h+Noc	77	68	81	75.33333	6.658328	0.000161

Main Point 4: TBC1D3C induces the decoupling of actin filament assembly and microtubule disassembly via relocating GEFH1 to the cytosol.
 With this, the authors show that induction of TBC1D3C expression does in fact not affect the microtubule network while affecting GEFH1 localization. The same criticism as previously mentioned applies to fluorescence images.

Thank you for your valuable criticism. We acknowledge that our previous version lacked quantitative analysis of the changes in GEF-H1 localization in relation to microtubule network. To address this, we have now conducted a colocalization analysis and calculated the correlation coefficients between GEF-H1 and microtubules as shown in Figures 4e-f. The quantification results provide a more reliable measurement of the changes in GEF-H1 distribution in relation to the microtubule

network. The associated quantification data have been included in the Source Data, which we have provided as a Supplemental File. Figure 4c-e and Source data are shown as below.

Fig.4d

Correlation coefficient	rep1	rep2	rep3	rep4	rep5	rep6	rep7	mean	SD	p value
Dox 0h	0.6411	0.7	0.58	0.72	0.65	0.68	0.6	0.67055	0.041649	
Dox 24h	0.4836	0.45	0.4	0.5	0.42	0.48	0.44	0.4668	0.023759	4.47981E-06
Dox 0h+Noc	0.42	0.43	0.4	0.45	0.5	0.42	0.52	0.425	0.007071	4.48472E-06
Dox 24h+Noc	0.549	0.5	0.52	0.44	0.56	0.48	0.45	0.5245	0.034648	0.057298373
Dox 24h+PTX	0.3182	0.35	0.4	0.42	0.45	0.34	0.36	0.3341	0.022486	0.005962979

Fig.4e

Correlation coefficient	rep1	rep2	rep3	rep4	rep5	rep6	rep7	mean	SD	p value
Dox 0h	0.6079	0.62	0.7	0.64	0.6	0.71	0.66	0.61395	0.008556	
Dox 24h	0.5489	0.57	0.62	0.7	0.61	0.65	0.73	0.55945	0.01492	0.611189688
Dox 0h+Noc	0.497	0.48	0.5	0.44	0.46	0.52	0.43	0.4885	0.012021	2.46027E-06
Dox 24h+Noc	0.55	0.5	0.44	0.52	0.46	0.53	0.47	0.525	0.035355	0.000515429

Main Point 5. Direct interaction between TBC1D3C and GEF-H1 Looks good to me.

Thank you for your positive feedback. We confirmed the direct interaction between TBC1D3C and GEF-H1 and further identified the specific interaction domains in both proteins through mutational truncation analysis.

Main Point 6: Luan et al. used SIM imaging to assess the co-localization of GEFH1 with microtubules before and after Dox treatment.

The treatment condition without doxycycline induction displays clear co-localization of GEFH1, whereas there seems to be less co-localization in the dox-induced treatment. What, however, needs to be taken into account is that the images don't

seem to be properly aligned in the dox condition, which becomes apparent when looking at the bottom edge of the image as well as the circle around the nucleus in the microtubule and GEFH1 condition. The authors should perform image registration, or at least manually align the separate images to properly assess the co-localization of the two channels.

Thank you for your valuable feedback. We acknowledge the misalignment issue in our initial images and have corrected this in the revised figures. Additionally, we have conducted a thorough analysis of the co-localization between GEF-H1 and microtubules by calculating their correlation coefficients and quantified F-actin stress fibers. The associated quantification data are now provided in the Source Data section of the Supplemental File. We appreciate your attention to this detail. Figure 6 and Source data are shown as below.

Fig.6b									
Correlation coefficient									
	rep1	rep2	rep3	rep4	rep5		mean	SD	p value
Dox 0h	0.7135	0.68	0.65	0.72	0.7		0.69675	0.023688	
Dox 24h	0.2959	0.33	0.35	0.4	0.2		0.31295	0.024112	0.000112
Fig.6c									
Relative intensity									
	rep1	rep2	rep3	rep4	rep5		mean	SD	p value
Dox 0h	1.259491	1.149852	0.746195	0.595526	1.248936		1	0.30813	
Dox 24h	1.80224	2.446798	1.762232	2.542988	2.253396		2.161531	0.361889	0.000652
Fig.6e									
Relative intensity									
	rep1	rep2	rep3	rep4		mean	SD	p value	
Dox 0 min	0.891097	1.16461	1.229082	0.71521		1	0.239819		
Dox 60 min	1.135564	1.476703	1.366022	0.998185		1.244118	0.216967	0.182345	
Dox 120 min	1.140332	1.310791	1.311305	0.964151		1.181645	0.165832	0.264694	
Dox 240 min	1.361328	1.575209	1.200183	0.854864		1.247896	0.303727	0.123715	
Dox 360 min	2.318071	2.267168	1.369836	1.111506		1.766645	0.616781	0.029848	
Dox 480 min	2.043092	2.817053	1.99835	1.711048		2.142386	0.473228	0.002528	
Dox 600 min	2.66295	1.683543	1.659411	1.204878		1.802696	0.614316	0.02543	
Dox 720 min	1.565087	2.587183	2.495938	1.93131		2.144879	0.483269	0.002707	

While the live-cell imaging does provide some additional proof that GEFH1 dissociates from the microtubules and the microtubules stay intact all the while, however, when looking at the panels in figure 6D, for this reviewer, it is impossible to tell by eye whether some z-drift occurred as it appears that the nucleus is clearly visible in the first timepoints in the actin channel and while it stays visible in the other two channels the nucleus does not appear to be visible in the actin channel towards the end of the timelapse. I'd double-check that the correct planes are shown, and no z-drift occurred. The corresponding movies are also a bit unclear as they appear to show a z-stack and time combined in one; this is a bit confusing, and I would suggest separating the different planes below each other, which can be easily done using open-source software such as ImageJ. Additionally, the 3 channels should also be visible in the same movie as they clearly correspond to each other, and this makes direct comparison of the different channels very difficult.

Thank you for your valuable suggestions. We really appreciate your detailed observations regarding our live-cell imaging data in Figure 6d. We have carefully reviewed the videos to ensure that no z-drift occurred and confirmed that the correct focal planes are shown throughout the timelapse.

To address your concerns about clarity and facilitate a more direct comparison between channels, we have now merged the three channels into a single movie. This merged presentation allows for a clearer and more straightforward visualization of the dynamics of GEF-H1 dissociation from the microtubules while ensuring that the microtubules and nuclei are visible and comparable across all time points.

General comments: Overall, what this reviewer finds mostly lacking are quantifications of various fluorescence images that would clearly strengthen the message of this paper. The western-blot data seems very clear and supportive of the main messages the authors want to make. The fluorescence data presented in Figure 6A (SIM imaging) needs to be aligned properly to correctly present colocalization data. The Live-cell imaging data needs to be double checked for corresponding image z-planes and the movies should be formatted to present the data more clearly. As an additional comment, the colormaps chosen to represent the various fluorescence images are not ideal. As a rule, I would suggest switching out the colormaps to green, cyan, and magenta since these are easier to see and provide more contrast, especially when printed (where the blue color would essentially disappear). I would also suggest to the authors to include scale bars that are of higher resolution since the ones in figure 6a are barely legible, and the ones in figure 6b are all but absent.

With the requested additional quantification of the IF data as well as the minor adjustments to the representation of the data I would recommend this for publication after a revision.

Thank you for your invaluable feedback. Following your suggestions, we have taken steps to address your concerns in this revised manuscript, which are summarized as follows for your convenient reference:

- 1) **Quantification and Statistical Analysis:** We have conducted extensive quantification of actin stress fibers, immunoblots, and colocalization coefficients across multiple figures where it is appropriate (Figures 1b-c, e-f, 2b-c, 3b-f, 4d-e, and 6b-e). The associated quantification data (in Excel) are provided in the Source Data as a Supplemental file.
- 2) **Alignment and Presentation of Fluorescence Data:** We have carefully realigned the SIM imaging data in Figure 6a, ensuring accuracy in colocalization. High-resolution scale bars have been added to Figure 6a.
- 3) **Live-cell Imaging Adjustments:** We have reviewed and adjusted the live-cell imaging data to ensure that the correct z-planes are shown consistently in Figure 6d. We have also merged the three channels into a single movie for a clearer and more straightforward visualization and comparison of the dynamics of GEF-H1, microtubules and F-actin stress fibers.
- 4) **Improvements to Visualization:** We have updated our images to green, cyan, and magenta, offering better visibility and contrast.
- 5) **Detailed Experimental Descriptions and Methodological Clarifications:** We have updated our Materials and Methods section with detailed descriptions of key procedures such as actin segmentation by ultracentrifugation, microtubule co-sedimentation assays, and peptide quantification by mass spectrometry. We have also clarified the CRISPR/Cas9-based generation of the GEF-H1 knockout stable cell line .

We believe that with these improvements we have improved and strengthened this

revised manuscript substantially. We really appreciate your feedback and hope that these revisions meet your expectations for publication.

September 9, 2024

Re: Life Science Alliance manuscript #LSA-2024-02585-TR-A

Dr. Zongping Xia
First Affiliated Hospital of Zhengzhou University
1st Jianshe Road, Zhengzhou, Henan
Zhengzhou 450052
China

Dear Dr. Xia,

Thank you for submitting your revised manuscript entitled "Decoupling actin assembly from microtubule disassembly by TBC1D3C-mediated direct GEF-H1 activation" to Life Science Alliance. The manuscript has been seen by the original reviewers whose comments are appended below. While the reviewers continue to be overall positive about the work in terms of its suitability for Life Science Alliance, some important issues remain.

Our general policy is that papers are considered through only one revision cycle; however, we are open to one additional short round of revision. Please note that I will expect to make a final decision without additional reviewer input upon re-submission.

Please submit the final revision within one month, along with a letter that includes a point by point response to the remaining reviewer comments.

To upload the revised version of your manuscript, please log in to your account: <https://lsa.msubmit.net/cgi-bin/main.plex>
You will be guided to complete the submission of your revised manuscript and to fill in all necessary information.

B. MANUSCRIPT ORGANIZATION AND FORMATTING:

Sincerely,

Reviewer #1 (Comments to the Authors (Required)):

The authors have largely addressed my concerns. Four points remain to be address, but none of these requires experiments.
1. A description of how cells with stress fibers were identified. Was there a particular algorithm in ImageJ that was used for

instance? If not, were they identified by inspection and, if so, how was bias avoided, e.g. was scoring done in a double blind manner.

2. In the pull down experiments in fig2b and c, the variation is very large. Ideally the experiment should be replicated a couple of more times to provide more confidence in the conclusions.

3. What algorithm was used to determine correlation coefficients and how was it implemented?

4. What is Noc? I assume nocodazole.

Reviewer #2 (Comments to the Authors (Required)):

This manuscript by Luan et al. describes the role of TBC1D3C in GEF-H1-mediated RhoA activation. The authors argue that actin and microtubule dynamics are decoupled and that TBC1D3C can have an important role in this process. I already stated in my original report that the manuscript certainly has merits. I find that the revised version is much improved and that the data provided support the conclusions from the article. It is well described why the authors started to study TBC1D3C, even if it is only expressed in hominoids and might not be a global factor in the communication between actin and microtubule assembly. I originally stated that the study is in most aspects well done but there are several weak points that will need some input from the authors.

Detailed comments:

1. line 123 and Fig.1B. The authors describe the increase in F-actin and a corresponding decrease. It is not described in the Mat&Meth section.

My response: the authors have described this well in the revised article

2. line 132. The authors claim that they have made an unbiased affinity purification. I do not agree, since there is a bias for the Rho subgroup. What about Rac and Cdc42? Alterations in their activities could also affect the status of actin.

My response: The authors have clarified this point.

3. Line 162, the authors found an accumulation of GEF-H1 but it is not clear if there were other RhoGEFs in the precipitate and if GEF-H1 was cherry picked because it followed the hypothesis.

My response: The authors have clarified why GEF-H1 was the most relevant candidate

4. Line 171, It is not clear what the authors mean by GEF-H1 activation? Activation in what sense? Increased binding?

My response: This is clear now

5. Extended data figures does that mean supplementary figures? This is a bit confusing.

My response: This is clear

6. Fig.3B and the panels showing cells onward. Here is the problem with showing single cells. How do we know that they are representative?

My response: Quantifications have been included in the revised manuscript.

7. Fig 4B. The microtubule staining is very difficult to follow in these minuscule sub panels. Would it be possible to show magnified section of some of the key conditions?

My response: This has been improved

8. Fig 4C. Again, the low resolution makes it difficult to evaluate the alleged effects on GEF-H1 relocalization. I suggest that the authors perform co-localization experiments in a more quantitative manner.

My response: This is sorted out

9. 6A the superresolution experiments are not really convincing. I suggest that the authors use another color than blue for microtubules (in 6A and B).

My response: This has been altered in the revised manuscript.

10. Fig. 6C It is not clear how the sedimentation assay has been done. Centrifugation, how?

My response: This is better explained in the revised text.

11. Fig.5E I suggest that the authors also include a figure of the domain organization of TBC1D3C. How many amino acid residues in GEF-H1 and TBC1D3C?

My response: This has been included.

Summary:

I think that the revised manuscript is much improved and that it is ready for publication in LSA.

Reviewer #3 (Comments to the Authors (Required)):

In the revised version of their manuscript, Luan et al. ("Decoupling actin assembly from microtubule disassembly by TBC1D3C-mediated direct GEF-H1 activation") have addressed several concerns raised by the reviewers. They incorporated additional quantifications of both fluorescence images and immunoblots, improved the alignment and presentation of fluorescence images, updated the accompanying movies, and expanded the methods section. However, several concerns have either not been fully addressed or have been introduced by the newly added quantifications.

I will in the following respond to the author's comments regarding my previous points made.

Main Point 1 response:

Luan et al. performed quantifications of stress fibres in fluorescence images of cells stained with phalloidin. This data supports

their hypothesis and enhances the reliability of the figure. However, the authors do not describe how they assessed the percentage of stress fibres, or the classification schemes used to determine whether cells exhibit stress fibres. In my experience, this assessment is rarely as straightforward as presented, as there is often a continuum to the degree to which a cell develops stress fibres. I would expect at least a description of the methodology and access to the raw data, including the number of cells quantified, rather than just the summary statistics provided in the supplementary data. Otherwise, properly assessing the presented quantification's validity is difficult.

Main Point 2 response:

While the authors have quantified immunoblots to support their results, they have not quantified the fluorescence images in this instance figure 2f-g, which would be necessary to provide quantitative insight into stress-fibre formation. Relying on visual assessment alone makes it difficult to fully evaluate the authors' interpretation based on the examples presented here. Quantification of the fluorescence images would strengthen the conclusions and provide clearer evidence.

Main Point 3 response:

The quantification of stress fibres faces the same issues as mentioned earlier: a lack of raw source data and a missing description of the methodology. Additionally, the authors seem to have misinterpreted my previous comment regarding the GEFH1-KO images.

My reference was not about generating a stable knockout cell line, as that is expected for a knockout. In subfigure H, the second-column panel at the 24-hour timepoint shows GEFH1 wild-type cells, with more than one cell visible in the actin panels (middle row). However, not all cells show fluorescence in the GEFH1 channel (upper row), or at least this is not apparent with the contrast scaling presented. My original comment referred to the fact that, since not all cells here show a visible GEFH1 signal, the authors may have used a cell line transiently expressing a tagged GEFH1 construct, rather than showing staining or stable expression of a tagged variant.

Given the assumption of transient expression, presenting GEFH1 fluorescence to showcase a knockout, as done in the uppermost row of panel H, is not informative, especially since only a small field of view with few cells is shown and no quantification has been performed. The absence of fluorescence could simply be due to the presented cells not expressing the marker.

If the authors are showing a stable construct or staining rather than a transient expression, this raises concerns about the reliability of the staining or construct expression. In such a case, the data might not be valid for assessing knockout efficiency or phenotyping.

If my interpretation is incorrect, I hope the authors will clarify the data presentation and figure description. Additionally, it is difficult to visually assess stress fibre formation from this single image.

On a separate note, it appears that the actin and merge images for the GEFH1 knockout in subfigure H have been accidentally swapped. This can be observed when looking at the green channel alone and increasing the contrast, where the GEFH1 and actin signals are identical. Although this doesn't significantly affect the data interpretation, it is an error that should be corrected to ensure proper data representation.

Main Point 4 response:

I appreciate that the authors have performed this quantification and I agree that it indeed adds value to the figure. I would like the authors to add detailed descriptions of the methodology or tools used to quantify this as I feel is standard practice.

Main Point 5 response:

Nothing to add.

Main Point 6 response:

The authors have made the suggested corrections regarding the misalignment issue. However, as mentioned previously, a detailed description of the methodology used to assess the correlation between the channels is required for the proper interpretation of the presented results.

Regarding subfigure D, which presents confocal imaging of GEFH1, microtubules, and actin, I still believe there is evidence of z-drift, particularly in the actin channel. The nucleus is clearly visible at the 0-hour time point but disappears at later time points, which, in my experience, is a strong indicator of z-drift.

I also appreciate the authors' response to my previous comments about the provided movies. However, I believe further improvements could be made. Specifically, the movies lack a scale bar and timestamp, which are essential for proper interpretation. Additionally, in the composite movie, the nuclei in the two channels seem to be not aligned. This potential misalignment could either be due to a delay introduced between the channels or as noted above, a z-drift, particularly in the actin channel. This issue should be addressed to ensure accurate data representation.

In Summary, I still find several points that should be addressed:

1. Lack of methodology and raw data: The quantification of stress fibres lacks a detailed description of how it was performed, as well as access to the true raw data, which is necessary for assessing the validity of the results. Similarly, the quantification of fluorescence correlation lacks methodology and description.
2. Additional need for fluorescence image quantification: While the authors have quantified most of the fluorescence images in some way or another, some panels still miss appropriate quantifications. Relying on visual assessment alone is insufficient in my opinion.
3. GEFH1-KO imaging: The way the imaging data is presented here does not provide reliable insight into the knockout efficiency and effect of the knockout.
4. Switched merge and Actin panels in Figure 3H: There is a potential error in the placement of GEFH1 and actin images in subfigure H that needs correction to ensure proper data representation.

5. Potential z-drift in confocal imaging Figure 6D: I find it difficult to assess the here presented timelapse data, particularly in the actin channel due to the apparent changes in the visibility of the nucleus during the timelapse which could indicate z-drift and thus directly influence the visibility of stress-fibres.
 6. Movies require improvement: The provided movies lack scale bars and timestamps, and there potentially is a misalignment between channels, which could occur due to either a temporal shift or z-drift. These issues should be addressed for clearer interpretation.
 7. General lack of scalebars: For proper interpretation of imaging data a standard requirement for figures should be the inclusion of scalebars in all images.
- Addressing these concerns would improve the transparency, accuracy, and robustness of the presented data.

Reviewer #1 (Comments to the Authors (Required)):

The authors have largely addressed my concerns. Four points remain to be address, but none of these requires experiments.

We really appreciate your constructive feedback on our manuscript. We are grateful for the opportunity to address your valuable critiques and hope that you are satisfied with this revised manuscript.

Please find our detailed responses in the following paragraphs.

1. A description of how cells with stress fibers were identified. Was there a particular algorithm in ImageJ that was used for instance? If not, were they identified by inspection and, if so, how was bias avoided, e.g. was scoring done in a double-blind manner.

We appreciate your constructive feedback. Stress fiber formation was quantified through manual inspection, where cells exhibiting lateral stress fibers were counted. A stress fiber was defined as an actin filament spanning the lateral width of the cell; if a cell displayed one or more stress fibers, it was classified as positive, and if not, as negative. Additionally, we measured the level of stress fibers through F-actin sedimentation, which further supported our hypothesis.

2. In the pull down experiments in fig2b and c, the variation is very large. Ideally the experiment should be replicated a couple of more times to provide more confidence in the conclusions.

Thank you for your valuable feedback. We appreciate your suggestion regarding the replication of our pull-down experiments. We would like to clarify that we observed a significant accumulation of active RhoA in both independent experiments. The observed variation in fold change reflects the inherent biological variability between the experiments. Despite this variability, we are confident in the conclusions drawn from our current data. We believe these findings contribute meaningfully to our understanding of the system.

3. What algorithm was used to determine correlation coefficients and how was it implemented?

Thank you for your insightful feedback. To determine the correlation coefficients,

we utilized ImageJ for image processing, which offers several pre-installed plugins, including a colocalization analysis procedure. Specifically, we employed Pearson's correlation coefficients to assess colocalization parameters in our study. This approach allows for robust analysis of the spatial relationship between the analyzed images.

4. What is Noc? I assume nocodazole.

Thank you for your insightful feedback. We appreciate your attention to detail. 'Noc' indeed refers to nocodazole throughout the manuscript, and we apologize for any confusion caused by the labeling. We will ensure clearer terminology in the revised version.

Reviewer #2 (Comments to the Authors (Required)):

This manuscript by Luan et al. describes the role of TBC1D3C in GEF-H1-mediated RhoA activation. The authors argue that actin and microtubule dynamics are decoupled and that TBC1D3C can have an important role in this process. I already stated in my original report that the manuscript certainly has merits. I find that the revised version is much improved and that the data provided support the conclusions from the article. It is well described why the authors started to study TBC1D3C, even if it is only expressed in hominoids and might not be a global factor in the communication between actin and microtubule assembly. I originally stated that the study is in most aspects well done but there are several weak points that will need some input from the authors.

Thank you for your thoughtful feedback and constructive suggestions, which have significantly enhanced the quality of our manuscript. We appreciate your recognition of the merits of our study and the improvements made in the revised version. Your insights have been invaluable in helping us clarify the objectives of our research.

We have undertaken substantial revisions to address your critiques and enhance the overall presentation of our findings. We are grateful for the opportunity to respond to your comments and hope that the revisions meet your expectations.

Below, you will find our detailed responses to each of your points.

Detailed comments:

1. line 123 and Fig.1B. The authors describe the increase in F-actin and a corresponding decrease. It is not described in the Mat&Meth section.
My response: the authors have described this well in the revised article.

Thank you for your insightful suggestion. We appreciate your attention to detail and have ensured that the description of the increase in F-actin, as well as its corresponding results, has been clearly included in the Materials and Methods section of the revised manuscript.

2. line 132. The authors claim that they have made an unbiased affinity purification. I do not agree, since there is a bias for the Rho subgroup. What about Rac and Cdc42? Alterations in their activities could also affect the status of actin.
My response: The authors have clarified this point.

Thank you very much for your valuable comments. We really appreciate your valuable comments.

3. Line 162, the authors found an accumulation of GEF-H1 but it is not clear if there were other RhoGEFs in the precipitate and if GEF-H1 was cherry picked because it followed the hypothesis.
My response: The authors have clarified why GEF-H1 was the most relevant candidate

Thank you very much for your valuable comments. We are glad to clarify this point.

4. Line 171, It is not clear what the authors mean by GEF-H1 activation? Activation in what sense? Increased binding?
My response: This is clear now

Thank you very much for your valuable comments. We are glad to address your concerns.

5. Extended data figures does that mean supplementary figures? This is a bit confusing.
My response: This is clear

Thank you very much for your valuable comments. We are glad to address your concerns.

6. Fig.3B and the panels showing cells onward. Here is the problem with showing single cells. How do we know that they are representative?
My response: Quantifications have been included in the revised manuscript.

Thank you very much for your valuable comments. We are grateful for your suggestion to include a quantification analysis which has strengthened our conclusions.

7. Fig 4B. The microtubule staining is very difficult to follow in these minuscule sub

panels. Would it be possible to show magnified section of some of the key conditions?

My response: This has been improved

Thank you very much for your valuable comments. We agree that a magnified section of some pictures really make it clearer to follow.

8. Fig 4C. Again, the low resolution makes it difficult to evaluate the alleged effects on GEF-H1 relocalization. I suggest that the authors perform co-localization experiments in a more quantitative manner.

My response: This is sorted out

Thank you very much for your valuable comments. We are grateful for your suggestion to include a quantification analysis which has strengthened our conclusions.

9. 6A the superresolution experiments are not really convincing. I suggest that the authors use another color than blue for microtubules (in 6A and B).

My response: This has been altered in the revised manuscript.

Thank you very much for your valuable comments. We agree that another color other than blue in immunofluorescent images is clearer to follow.

10. Fig. 6C It is not clear how the sedimentation assay has been done. Centrifugation, how?

My response: This is better explained in the revised text.

Thank you very much for your valuable suggestion, which is important to perfect the manuscript.

11. Fig.5E I suggest that the authors also include a figure of the domain organization of TBC1D3C. How many amino acid residues in GEF-H1 and TBC1D3C?

My response: This has been included.

Thank you very much for your valuable suggestion, which is important to perfect

the manuscript.

Summary:

I think that the revised manuscript is much improved and that it is ready for publication in LSA.

We sincerely appreciate your positive feedback on our revised manuscript. Your insightful comments have significantly enhanced the quality of our work, and we are grateful for your guidance throughout the revision process.

Reviewer #3 (Comments to the Authors (Required)):

In the revised version of their manuscript, Luan et al. ("Decoupling actin assembly from microtubule disassembly by TBC1D3C-mediated direct GEF-H1 activation") have addressed several concerns raised by the reviewers. They incorporated additional quantifications of both fluorescence images and immunoblots, improved the alignment and presentation of fluorescence images, updated the accompanying movies, and expanded the methods section. However, several concerns have either not been fully addressed or have been introduced by the newly added quantifications.

I will in the following respond to the author's comments regarding my previous points made.

We sincerely appreciate your insightful and constructive feedback on our manuscript. Your thorough review has been invaluable, and we have made significant revisions to enhance the clarity and accuracy of our work in response to your critiques. We trust that these revisions have addressed your concerns and met with your expectations. Below, please find our detailed responses to each of your comments."

Main Point 1 response:

Luan et al. performed quantifications of stress fibres in fluorescence images of cells stained with phalloidin. This data supports their hypothesis and enhances the reliability of the figure. However, the authors do not describe how they assessed the percentage of stress fibres, or the classification schemes used to determine whether cells exhibit stress fibres. In my experience, this assessment is rarely as straightforward as presented, as there is often a continuum to the degree to which a cell develops stress fibres. I would expect at least a description of the methodology and access to the raw data, including the number of cells quantified, rather than just the summary statistics provided in the supplementary data. Otherwise, properly assessing the presented quantification's validity is difficult.

Thank you for your valuable feedback. We apologize for not providing a detailed classification scheme for stress fiber quantification. To clarify, we quantified stress fiber formation by manually counting cells that displayed lateral stress fibers. A stress fiber was defined as an actin filament spanning the lateral width of the cell; cells exhibiting one or more stress fibers were categorized as positive, while those without were marked as negative. All analyses in Figure 1a were performed based on three independent experiments. For Figure 1b, the cell counts were as follows: Control: n = 95 cells; Dox 6 h: n = 110 cells; Dox 12 h: n = 98 cells; Dox 24 h: n = 98 cells; Dox 24 h + CytoD: n = 101 cells; Dox 24 h + Pha: n = 104 cells. In Figure 1c, the counts were: Control: n = 56 cells; Dox 6 h: n = 60 cells; Dox 12 h: n = 58 cells; Dox 24 h: n

= 67 cells; Dox 24 h + CytoD: n = 63 cells; Dox 24 h + Pha: n = 54 cells.

We appreciate your request for raw data. We have uploaded the raw data for Figures 1 and 3 to the journal's website along with this submission for easy access and independent verification. If you would like further raw image data, we would be more than happy to provide it directly via email. Please feel free to contact us at zxia2018@zzu.edu.cn.

Main Point 2 response:

While the authors have quantified immunoblots to support their results, they have not quantified the fluorescence images in this instance figure2f-g, which would be necessary to provide quantitative insight into stress-fibre formation. Relying on visual assessment alone makes it difficult to fully evaluate the authors' interpretation based on the examples presented here. Quantification of the fluorescence images would strengthen the conclusions and provide clearer evidence.

We appreciate your valuable feedback regarding the quantification of actin stress fibers. We agree with you about the importance of this kind of analysis in supporting our conclusions. Following your suggestion, we have conducted quantifications for Figures 2f and 2h (previously Figures 2f-g) by analyzing a minimum of 50 cells per experiment to determine the percentage of cells exhibiting actin stress fiber, with the results now presented in Figures 2g and 2i. For your convenience, we have included the updated Figures 2f-i below.

All quantified data (n=3 from three biological replicates) have been included in the Source Data, provided as a Supplemental File.

Main Point 3 response:

The quantification of stress fibres faces the same issues as mentioned earlier: a lack of raw source data and a missing description of the methodology. Additionally, the authors seem to have misinterpreted my previous comment regarding the GEFH1-KO images.

My reference was not about generating a stable knockout cell line, as that is expected for a knockout. In subfigure H, the second-column panel at the 24-hour timepoint shows GEFH1 wild-type cells, with more than one cell visible in the actin panels (middle row). However, not all cells show fluorescence in the GEFH1 channel (upper row), or at least this is not apparent with the contrast scaling presented. My original comment referred to the fact that, since not all cells here show a visible GEFH1 signal, the authors may have used a cell line transiently expressing a tagged GEFH1 construct, rather than showing staining or stable expression of a tagged variant.

Given the assumption of transient expression, presenting GEFH1 fluorescence to showcase a knockout, as done in the uppermost row of panel H, is not informative, especially since only a small field of view with few cells is shown and no quantification has been performed. The absence of fluorescence could simply be due to the presented cells not expressing the marker.

If the authors are showing a stable construct or staining rather than a transient expression, this raises concerns about the reliability of the staining or construct expression. In such a case, the data might not be valid for assessing knockout efficiency or phenotyping.

If my interpretation is incorrect, I hope the authors will clarify the data presentation and figure description. Additionally, it is difficult to visually assess stress fibre formation from this single image.

On a separate note, it appears that the actin and merge images for the GEFH1 knockout in subfigure H have been accidentally swapped. This can be observed when looking at the green channel alone and increasing the contrast, where the GEFH1 and actin signals are identical. Although this doesn't significantly affect the data interpretation, it is an error that should be corrected to ensure proper data representation.

Thank you for your valuable feedback and for pointing out the issues with our data presentation. We apologize for our earlier misinterpretation of your comment regarding the GEF-H1-KO images. To address your concerns and provide greater clarity, we have repeated the experiments and replaced panel h with new images. This updated panel now shows fields of views with more cells to better demonstrate the expression and knockout of GEF-H1 and actin stress fiber formation in response to TBC1D3C expression. We have placed panel h below for your convenient reference.

h

Main Point 4 response:

I appreciate that the authors have performed this quantification and I agree that it indeed adds value to the figure. I would like the authors to add detailed descriptions of the methodology or tools used to quantify this as I feel is standard practice.

Thank you for your valuable feedback and for acknowledging the improvement in the quantification. We apologize for lacking detailed descriptions of the methodology or tools for the quantification. Following your suggestions, we have added a description of the methodology under “Quantitative image analysis” in the “Materials and Methods” section of the revised manuscript.

Specifically, we utilized ImageJ for image processing, which includes several pre-installed plugins for colocalization analysis. In this study, we calculated Pearson's correlation coefficients, which provide a robust quantitative measure of colocalization.

Main Point 5 response:

Nothing to add.

Thank you very much for your valuable suggestion, which is important to improve the manuscript.

Main Point 6 response:

The authors have made the suggested corrections regarding the misalignment issue. However, as mentioned previously, a detailed description of the methodology used to assess the correlation between the channels is required for the proper interpretation of the presented results.

Thank you for your valuable feedback. We apologize for not providing a more detailed description of methodology in our analysis. We have now included a detailed

description of the methodology in this revised manuscript. Specifically, for the measurement of correlation coefficients in Fig. 6a, we analyzed high-resolution images using the ImageJ software with its pre-installed plugins, one of which facilitates colocalization analysis through colocalization indices. We calculated Pearson's correlation coefficients to quantitatively assess colocalization.

Regarding subfigure D, which presents confocal imaging of GEFH1, microtubules, and actin, I still believe there is evidence of z-drift, particularly in the actin channel. The nucleus is clearly visible at the 0-hour time point but disappears at later time points, which, in my experience, is a strong indicator of z-drift.

I also appreciate the authors' response to my previous comments about the provided movies. However, I believe further improvements could be made. Specifically, the movies lack a scale bar and timestamp, which are essential for proper interpretation. Additionally, in the composite movie, the nuclei in the two channels seem to be not aligned. This potential misalignment could either be due to a delay introduced between the channels or as noted above, a z-drift, particularly in the actin channel. This issue should be addressed to ensure accurate data representation.

Thank you for your valuable feedback regarding subfigure d and the movies. We really appreciate your careful observations. We have made further improvement for panel d, which is also shown below for your reference.

We agree with you that there could be a possibility of Z-drift, especially in the actin channel during the long-term live-cell imaging. This drift could occur due to subtle focus shifts over extended imaging sessions, where cell growth, movements and cytoskeleton dynamics might cause slight changes in focal plane. Actin, as a highly dynamic component in our analysis, might be more susceptible to these focus fluctuations, which might well explain the visibility of the nucleus at the 0-hour time point and its disappearance at later time points.

As to the movies, we have now included a scale bar and timestamp to improve clarity and facilitate better interpretation. We acknowledge the misalignment and the possibility of a slight delay or Z-drift affecting the actin channel. With the current

dataset, we were unable to make further corrections to the existing movies. However, we have reviewed the data thoroughly and believe that despite these minor issues, the movies still provide valuable support to our conclusions. We have made every effort to optimize the presentation within the constraints of the data, and we hope that these improvements meet your expectations and effectively convey the key findings.

We sincerely appreciate your constructive feedback, which has led us to further refine our imaging analysis and data presentation.

In Summary, I still find several points that should be addressed:

1. Lack of methodology and raw data: The quantification of stress fibres lacks a detailed description of how it was performed, as well as access to the true raw data, which is necessary for assessing the validity of the results. Similarly, the quantification of fluorescence correlation lacks methodology and description.
2. Additional need for fluorescence image quantification: While the authors have quantified most of the fluorescence images in some way or another, some panels still miss appropriate quantifications. Relying on visual assessment alone is insufficient in my opinion.
3. GEF-H1-KO imaging: The way the imaging data is presented here does not provide reliable insight into the knockout efficiency and effect of the knockout.
4. Switched merge and Actin panels in Figure 3H: There is a potential error in the placement of GEFH1 and actin images in subfigure H that needs correction to ensure proper data representation.
5. Potential z-drift in confocal imaging Figure 6D: I find it difficult to assess the here presented timelapse data, particularly in the actin channel due to the apparent changes in the visibility of the nucleus during the timelapse which could indicate z-drift and thus directly influence the visibility of stress-fibres.
6. Movies require improvement: The provided movies lack scale bars and timestamps, and there potentially is a misalignment between channels, which could occur due to either a temporal shift or z-drift. These issues should be addressed for clearer interpretation.
7. General lack of scalebars: For proper interpretation of imaging data a standard requirement for figures should be the inclusion of scalebars in all images.

Addressing these concerns would improve the transparency, accuracy, and robustness of the presented data.

We would like to express our sincere gratitude to you for your comprehensive review and constructive feedback on our manuscript. We have carefully considered each of your comments and have made significant efforts to address your concerns in the revised version. Below, we outline the steps we have taken to enhance the manuscript:

1. Quantification, Statistical Analysis and Raw Data Availability: We have now included a detailed description of the methodology used for the quantification of actin stress fibers in the revised manuscript. We have also uploaded the raw data

for Figures 1 and 3 to the journal's website along with this submission for easy access and independent verification. If you would like further raw image data, we would be more than happy to provide it directly via email. Please feel free to contact us at zxia2018@zzu.edu.cn. We hope these additions should allow for a clearer assessment of our results.

2. **Additional Fluorescence Image Quantification:** We conducted thorough quantifications of actin stress fibers in Figures 2f-i, analyzing at least 50 cells per experiment to determine the percentage of cells exhibiting actin stress fibers.
3. **GEF-H1-KO Imaging Improvement:** We apologize for our earlier misinterpretation of your comment regarding the GEF-H1-KO images. To address your concerns and provide greater clarity, we have repeated the experiments and replaced Panel h with new images. This updated panel now shows fields of views with more cells to better demonstrate the expression and knockout of GEF-H1 and actin stress fiber formation in response to TBC1D3C expression.
4. **Corrected Imaging Panels:** We have updated Figure 3H as described in point 3. This update has provided a more reliable presentation of the data.
5. **Addressing Potential Z-Drift in Confocal Imaging (Figure 6D):** We have replaced the time-lapse images to better illustrate the visibility of the nucleus. We agree with you that there is indeed a possibility of Z-drift, especially in the actin channel during long-term live-cell imaging, due to subtle focus shifts over extended imaging sessions, where cell growth, movements and cytoskeleton dynamics might cause slight changes in focal plane. However, given the limitations of the current dataset, further correction of the Z-drift in this time-lapse data is not feasible. Despite these challenges, we believe that the overall conclusions drawn from these data remain robust, and the images are sufficient to support our findings. We hope that this explanation clarifies our efforts to address this issue and reassures you of the reliability of our results.
6. **Improvements to Movies:** We have now added scale bars and timestamps to the movies. We also noted potential misalignment between channels in the last few intervals, which may result from temporal shifts or Z-drift. Due to the constraints of the existing dataset, we were unable to make additional corrections to the current movies. However, we have carefully reviewed the data, and despite these minor limitations, we believe that the movies still adequately illustrate the key findings of our study. We have made every effort to enhance the presentation quality and hope that these improvements meet with your expectations.
7. **General Inclusion of Scale Bars in All Images:** We have carefully examined and included scale bars in all the images to facilitate proper interpretation.

We believe these revisions significantly enhance the transparency, accuracy, and robustness of our findings. We sincerely appreciate your feedback and hope that these changes meet with your expectations for publication.

October 10, 2024

RE: Life Science Alliance Manuscript #LSA-2024-02585-TRR

Dr. Zongping Xia
First Affiliated Hospital of Zhengzhou University
1st Jianshe Road, Zhengzhou, Henan
Zhengzhou 450052
China

Dear Dr. Xia,

Thank you for submitting your revised manuscript entitled "Decoupling actin assembly from microtubule disassembly by TBC1D3C-mediated direct GEF-H1 activation". We would be happy to publish your paper in Life Science Alliance pending final revisions necessary to meet our formatting guidelines.

- please be sure that the authorship listing and order is correct
- please add a category for your manuscript to our system
- please add a figure callout for your Figure S2 to your main manuscript text
- please add sizes next to all blots

LSA now encourages authors to provide a 30-60 second video where the study is briefly explained. We will use these videos on social media to promote the published paper and the presenting author (for examples, see <https://docs.google.com/document/d/1-UWCfbE4pGcDdcgzcmiuJl2XMBJnxKYeqRvLLrLSo8s/edit?usp=sharing>). Corresponding or first-authors are welcome to submit the video. Please submit only one video per manuscript. The video can be emailed to contact@life-science-alliance.org

A. FINAL FILES:

B. MANUSCRIPT ORGANIZATION AND FORMATTING:

Sincerely,

October 16, 2024

RE: Life Science Alliance Manuscript #LSA-2024-02585-TRRR

Dr. Zongping Xia
First Affiliated Hospital of Zhengzhou University
1st Jianshe Road, Zhengzhou, Henan
Zhengzhou 450052
China

Dear Dr. Xia,

Thank you for submitting your Research Article entitled "Decoupling actin assembly from microtubule disassembly by TBC1D3C-mediated direct GEF-H1 activation". It is a pleasure to let you know that your manuscript is now accepted for publication in Life Science Alliance. Congratulations on this interesting work.

DISTRIBUTION OF MATERIALS:

Again, congratulations on a very nice paper. I hope you found the review process to be constructive and are pleased with how the manuscript was handled editorially. We look forward to future exciting submissions from your lab.

Sincerely,
